# On the Modeling Capabilities of Large Language Models for Sequential Decision Making

**Martin Klissarov**[*]
Mila, McGill University

**Devon Hjelm**
Apple

**Alexander Toshev**
Apple

**Bogdan Mazoure**
Apple

## Abstract

Large pretrained models are showing increasingly better performance in reasoning and planning tasks across different modalities, opening the possibility to leverage them for complex sequential decision making problems. In this paper, we investigate the capabilities of Large Language Models (LLMs) for reinforcement learning (RL) across a diversity of interactive domains. We evaluate their ability to produce decision-making policies, either directly, by generating actions, or indirectly, by first generating reward models to train an agent with RL. Our results show that, even without task-specific fine-tuning, LLMs excel at reward modeling. In particular, crafting rewards through artificial intelligence (AI) feedback yields the most generally applicable approach and can enhance performance by improving credit assignment and exploration. Finally, in environments with unfamiliar dynamics, we explore how fine-tuning LLMs with synthetic data can significantly improve their reward modeling capabilities while mitigating catastrophic forgetting, further broadening their utility in sequential decision-making tasks.

## 1 Introduction

Large Language Models (LLMs) are generative models of natural language that can produce accurate general and domain-specific knowledge (Singhal et al., 2022; Imani et al., 2023; Manigrasso et al., 2024; Liu et al., 2024a), reason over long textual contexts (Reid et al., 2024), and generalize zero-shot (Kojima et al., 2022). These capabilities suggest that LLMs might be well-suited for complex sequential decision-making problems, such as in embodied settings where an agent acts in an environment. Recent research has begun exploring this potential, investigating how LLMs can serve as sources of intrinsic motivation (Wang et al., 2024; Klissarov et al., 2024), demonstrating world modeling capabilities (Lin et al., 2024; Liu et al., 2024b), and for acting and/or planning directly in an environment (Wang et al., 2023; Padalkar et al., 2023; Zhang et al., 2024).

However, as the predominant paradigm for training LLMs is not inherently aligned with the challenges of sequential decision-making problems, such as active exploration, it is not obvious how to best bridge their capabilities to tackle such challenges in a general manner. We study this problem through the lens of reinforcement learning (RL, Sutton & Barto, 2018), which formalizes how an agent interacts with an environment, receiving scalar rewards for each of its actions over a trajectory. We examine the capabilities of LLMs to solve RL tasks by comparing how they model policies 1) directly by generating action tokens, to 2) indirectly through a reward model derived from the LLM to be used within an RL algorithm. We perform a comprehensive evaluation on a diverse set of domains, including MiniWob (Liu et al., 2018), NetHack (Küttler et al., 2020), and Wordle (Lokshtanov & Subercaseaux, 2022), and MetaWorld (Yu et al., 2019). The environments we study present a variety of challenges, such as different action space granularities, observation modalities ranging from natural language to pixel data, and varying horizon lengths.

We first consider the off-the-shelf capabilities of LLMs for decision-making without updating them through additional gradient updates coming from the RL task. We find that indirectly modeling policies by first extracting knowledge from LLMs in the form of a Bradley-Terry model (Bradley & Terry, 1952; Christiano et al., 2017) provides the best and most consistent performance across the environments we study. We empirically analyze the various benefits, and limitations, provided by

---

[*] Work done during an Apple internship. Correspondence to: `martin.klissarov@mail.mcgill.ca`.

this approach, showing that it improves on long-standing challenges in RL problems, such as credit assignment and exploration.

Finally, while LLMs possess knowledge useful for many decision making tasks of interest, domains with complex or unfamiliar dynamics can significantly restrict their broader utility. We explore how fine-tuning an LLM with domain-specific data can bridge this knowledge gap and study the effect of this procedure on the LLM's previous knowledge, as measured through success on datasets like POPE (Li et al., 2023b), GQA (Hudson & Manning, 2019), AI2D (Kembhavi et al., 2016) and MMMU (Yue et al., 2024). Our investigation reveals that fine-tuning for indirect policy modeling mitigates catastrophic forgetting more effectively than direct policy modeling, offering a broadly applicable strategy for leveraging LLMs across diverse sequential decision-making tasks.

## 2 USING LANGUAGE MODELS TO SOLVE RL TASKS

We first introduce the types of RL problems as well as formalize the methodologies for using LLMs for RL tasks used in this work.

**Reinforcement Learning.** An RL task can be defined through a Markov Decision Process (MDP, Puterman, 2014), which is composed of a state space $\mathcal{S}$, an action space $\mathcal{A}$, a transition function $p : \mathcal{S} \times \mathcal{A} \to \Delta(\mathcal{S})$ which describes the forward dynamics of the system, a reward function $r : \mathcal{S} \times \mathcal{A} \to \mathbb{R}$ and a discount factor $\gamma \in [0, 1]$. Since it is often the case that the state is only partially observable, we also assume the environment emits an observation $o_t \sim p_{\mathcal{O}} : \mathcal{S} \to \Delta(\mathcal{O})$ from observation space $\mathcal{O}$. A policy, or *actor*, is a probability distribution $\pi : \mathcal{S} \to \Delta(\mathcal{A})$ which describes the action to be taken at every step. The objective of a rational actor is to maximize the expected cumulative rewards over horizon $H > 0$,

$$\max_{\pi} \mathbb{E}[\sum_{t=0}^{H} \gamma^t r(s_t, \pi(s_t))|s_0] = \max_{\pi} \mathbb{E}_{s_0}[V^{\pi}(s_0)], \tag{1}$$

where the value function, $V^{\pi}(s)$, represents the expected discounted sum of rewards over the entire trajectory, re-weighted by the environment's dynamics model, $p$, and the actor's policy, $\pi$.

**Large Language Models.** An LLM is a generative model of discrete random variables (i.e. tokens) conditioned on a history (i.e. context). The LLM models the data distribution autoregressively:

$$p(x_{t+1}|x_1, .., x_t) = \prod_{t'=1}^{t} p(x_{t'}|x_{<t'}) = \texttt{LLM}(x_{<t}, l) \tag{2}$$

where $x \in \mathcal{X}$ are token variables taken from a valid vocabulary. The suitability of LLMs for solving RL tasks without additional fine-tuning primarily hinges on the hypothesis that LLMs contain information – i.e., *knowledge* – about the underlying MDP, for instance, through the policy or reward function. *How* that information is extracted depends on the data the LLM was trained on, the ability of the practitioner to properly prompt the model and interpret its responses to solve decision-making tasks.

### 2.1 PROMPTING

In this section, we describe the inputs, or *prompts*, to the LLM used in this work which allow to change the LLM's output distribution to be useful for solving RL tasks. All prompts in this work use 1) task specification using natural language as input to provide information about the MDP to the LLM as context and 2) episode history in order to address issues of partial-observability in some environments (similar to the Act-only baseline prompt found in Yao et al., 2022). We additionally use the following set of techniques,

- **Chain of Thought**. By prompting the LLM to provide a step-by-step reasoning process for its output, rather than just the final answer, we can help surface its internal decision-making and improve the resulting performance (Wei et al., 2022).

- **In-Context Learning**. To enhance the LLM's ability to solve the task, example solutions (e.g., from expert policies) are provided for in-context learning (Brown et al., 2020), where solutions contain sequences of a combination of states, actions, and rewards.

- **Self-Refinement**. To further refine its output, the LLM is prompted to provide recursive criticism and improvement from its generated outputs. This general strategy knows many variants, such as feedback from an environment (Yao et al., 2022), self-critique (Zelikman et al., 2022), or self-reflection (Shinn et al., 2023). In this work, we use Recursive Criticism and Improvement (RCI, Kim et al., 2024) for its state-of-the-art performance on web agent domains and general applicability. In its original form, the LLM is given a task description and generates a high-level plan. This plan is used along with the task description and current state to refine an action so that it is grounded in the current observation and the action space.

## 2.2 POLICY MODELING USING LLMS

As shown in Equation 1, the goal of a decision making agent is to learn a high performing policy $\pi$. This can be done either by maximizing the expected cumulative rewards and directly modeling the policy parameters (Sutton et al., 1999; Kakade & Langford, 2002). Equivalently, this can be done indirectly by first modeling the parameters of the value function and applying a greedy operator, such as in Q-Learning (Watkins & Dayan, 1992). A similar separation between direct and indirect approaches can be useful to study the capabilities of LLMs to model RL policies.

**Direct Policy Modeling.** The most straightforward way to obtain a policy using LLMs is for the LLM to generate tokens that will be directly interpreted as actions from the environment, $a \in \mathcal{A}$ (Yao et al., 2022; Shinn et al., 2023; Kim et al., 2024). To ensure the outputted actions adhere to the environment's action set, the LLM output tokens can be projected back onto $\mathcal{A}$ using projection operator $\text{proj}(\cdot, \mathcal{A})$ (e.g., see Huang et al., 2022; Kim et al., 2024, for examples of projection operators). A variety of prompting techniques can be combined to increase the ability of the LLM to act, without task-specific fine-tuning, as a policy, which we detail in Section 2.1. This direct policy method will be referred to in our experiments as **LLM Policy**.

**Indirect Policy Modeling.** On the other hand, we can prompt the LLM to output tokens representing intermediate quantities that will then be used to learn a policy. For example, one can model the forward dynamics of the environment for planning (Liu et al., 2024b) or an affordance model for action selection (Mullen Jr & Manocha, 2024).

In this work, we focus on the case where these intermediate quantities will be used to generate rewards – i.e., a **reward model** – which will then be maximized by an off-the-shelf RL policy. In Section 2.3, we enumerate the different approaches for modeling reward functions with LLMs covered in our work. It is important to note that there exists many more ways in which we could indirectly model the policy. In Appendix A.4, we present in detail these possibilities and, in Figure 2b, provide initial investigations that showcase their potential and limitations.

In direct policy modeling experiments (LLM Policy), we found combining all of the prompting techniques in Section 2.1 to work the best, while for indirect modeling methods through reward we relied only on chain-of-thought prompting. Additional details, such specific prompt details and ablations on these choices are presented in the Appendix A.3.

## 2.3 INDIRECTLY MODELING POLICIES THROUGH REWARD MODELS

We consider a diversity of methods for modeling reward functions using LLMs, with a particular attention to methods that are applicable to a diversity of environments and modalities. We study the following set,

- **Direct Scalar**. (Kwon et al., 2023) The LLM generates tokens that directly encode the reward (e.g., as a float or integer) given an observation (or a sequence of observations and actions). This reward is then given to the RL agent.

- **AI Feedback** (Lee et al., 2023; Klissarov et al., 2024)). Ask the LLM to express a preference $y = \{1, 2, \varnothing\}$ between two observations, $o_1$ and $o_2$, for the one showing the most progress towards a certain goal, or no preference if both observations are equally

good. These labels can then be collected as a dataset of observation-preference tuples $\mathcal{D}_{\text{pref}} = \{(o_1^{(i)}, o_2^{(i)}, y^{(i)})\}_{i=1}^M$, which are then used to train a reward function modeled as,

$$r_\theta = \arg\min_\theta \mathbb{E}_{(o_1, o_2, y) \sim \mathcal{D}_{\text{pref}}} \left[ \mathbb{I}[y=1] \log P_\theta[o_1 \succ o_2] + \mathbb{I}[y=2] P_\theta[o_2 \succ o_1] \right.$$
$$\left. + \frac{1}{2} \mathbb{I}[y=\varnothing] \log \left( P_\theta[o_1 \succ o_2] P[o_2 \succ o_1] \right) \right] \tag{3}$$

where $P_\theta[o_1 \succ o_2] = \frac{e^{r_\theta(o_1)}}{e^{r_\theta(o_1)} + e^{r_\theta(o_2)}}$ the probability of preferring an observation to another, referred to as the Bradley-Terry model for preference learning (Bradley & Terry, 1952). The minimization of this equation is commonly done through binary cross-entropy.

- **Reward as Code** (Xie et al. (2023),Yu et al. (2023); Ma et al. (2023)). Prompt the LLM to write code that will take as input a subset of symbolic features from the environment observations and will produce a scalar output representing the reward. The code defining the reward function is then updated throughout environment interactions as in Li et al. (2023a). When symbolic features are not available, these are constructed as in Venuto et al. (2024).

- **Embedding-based** (Rocamonde et al., 2023; Du et al., 2023; Liu et al., 2024b). Instead of querying language tokens from the LLM, we can instead, for a given input, leverage the information encoded in its latent represention, or embeddings. These embeddings are used to calculate the cosine similarity with respect to the embeddings of natural language specification of a goal or a behaviour. The resulting similarity value is given as a reward to the agent.

Additional details, such specific prompts, are presented in the Appendix A.2.

## 3 PERFORMANCE OF INDIRECT AND DIRECT POLICY MODELS

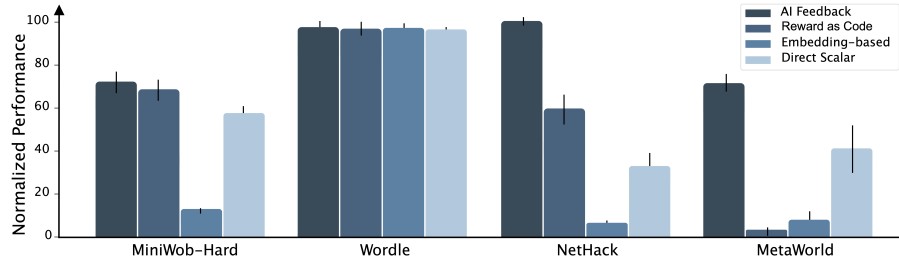

Figure 1: **AI feedback has the highest performance across different reward models derived from tested LLMs.** AI feedback, which is a preference-based method for deriving a reward model from an LLM, generally outperforms other methods.

Due to fundamentally different challenges between direct and indirect policy modeling approaches, conducting a fair comparison requires care. For example, using the LLM directly as a policy requires grounding its outputs in the action space defined by the environment (Ahn et al., 2022; Huang et al., 2022). As the action space can vary significantly between environments and attempting to solve this problem adds additional algorithm- or domain-specific complexities (e.g. by crafting skills, see (Ahn et al., 2022; Wang et al., 2023)), we fix our experimental setting to the following

1. **Atomic actions.** We only study approaches which can directly interface with the action space supported in the environment. In other words, the action space is at least a subspace of the space of language generated by the LLM. This allows for a more direct comparison across a variety of domains and study the relationship between an LLM's knowledge and the fixed action space defined by the environment.

2. **No finetuning.** In most of the paper we assume that LLMs are used without any gradient updates, i.e. *without fine-tuning* from the RL task, and evaluate their off-the-shelf capabilities. In Section 5, we perform a preliminary study on the trade-offs between fine-tuning for direct and indirect policy modeling.

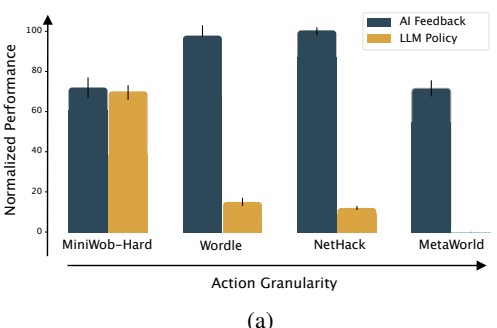 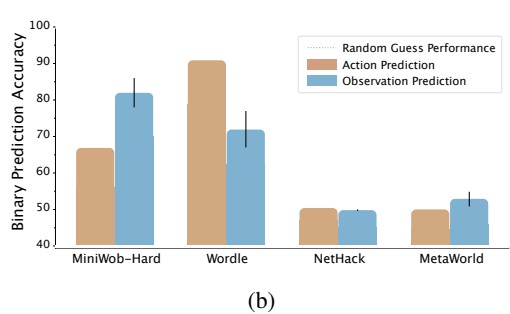

(a) (b)

Figure 2: **a) Building a reward model more-readily solves RL tasks than using an LLM as an actor.** LLM-policy only performs well in domains with coarse-grained actions while LLM feedback presents strong performance across the entire range of action granularities. **b) LLMs have unreliable zero-shot understanding of the environment dynamics.** While LLMs can be used to craft useful reward models, their failure as direct policies may be explained by their poor understanding of the action space and the transition function.

We investigate four separate domains, where each domain aims to highlight a specific capability of LLMs: 1) MiniWob-Hard, a subset of hard tasks from the full MiniWob suite, tests web interaction in observation/action spaces close to natural language, 2) Wordle measures reasoning and planning capabilities, 3) NetHack presents the difficulty of exploring open-ended environments under partial observability, long horizons and procedural scenarios, and 4) MetaWorld assesses the ability to control low-level, high-frequency actions in continuous space. We provide a detailed description of each domain in Appendix A.1.

Direct policy modeling is done by querying the closed source GPT-4o model, whereas indirect policy modeling is done through the open source models of Llama 3 (Dubey et al., 2024), when environment observations consist of text, and PaliGemma (Beyer et al., 2024), when environment observation consist of pixel images. In the Appendix A.11, we investigate a larger set of LLMs, including Claude 3.5, Gemini Pro 1.5, Llama 3.2-V and Qwen 2.5 (Qwen Team, 2024). All results are averaged over 10 seeds with error bars indicating the standard error.

**Indirect policy modeling through rewards.** We first present a comparison of the various indirect policy modeling approaches discussed in Section 2.3. In these experiments, the LLM generates a reward function which will be given to a RL agent for optimization, without access to any rewards coming from the environment. When learning policies through RL we do not perform any hyper-parameter search and simply borrow the existing empirical setup for each domain, as detailed in Appendix A.1.

In Figure 1, we present the performance across domains as measured by the average success rate on all domains, except for NetHack, where performance (the in-game score) is normalized by the highest recorded value. Results show that AI feedback is the only method that successfully crafts rewards across all environments and modalities . On easier domains such as MiniWob-Hard, which consists of short episodes and limited scope of variations, the Direct Scalar method performs nearly as well as AI feedback. However, the disparity between methods is much more pronounced on harder, open-ended tasks such as NetHack. Out of all the methods, Embedding-based leads to the lowest performance. Finally, the effectiveness of Reward as Code appears to be highly contingent on the availability of symbolic features for code processing. In Appendix A.5, we further examine the assumptions—such as access to functional knowledge of the environment—under which Reward as Code can achieve performance comparable to AI feedback.

**Direct vs indirect policy modeling.** We now compare the direct policy modeling method, LLM Policy, to the best performing indirect modeling method, AI feedback, reporting performance across the same set of domains. Results in Figure 2a show that, despite the more complex prompting

---

In Appendix A.6, we verify that AI feedback yields policies with performance on par with those optimized using human-designed environment rewards.

strategies and the use of a more capable closed source model, LLM Policy is unable to perform well in most environments, with the exception of MiniWob-Hard, where the performance is on-par with AI feedback. Given that the reward modeling baseline proceeds to fine-tuning the RL policy, we may wonder if, for a similar amount of compute and environment samples, what performance could the LLM Policy achieve by fine-tuning the LLM. In Appendix A.13, we also study this question, revealing that reward modeling is significantly more efficient. In Figure 15 of Appendix A.12, we additionally investigate whether the reward obtained by the AI feedback method can be given to the LLM Policy as context, in order to improve the direct modeling performance. Results indicate that this additional information does not significantly change the performance.

A question emerging from these results is: what factors cause this significant performance disparity between direct and indirect policy models? One possible explanation is that LLMs, when directly queried for actions in an unfamiliar environment, may struggle to understand its dynamics (e.g., the transition function and action space). To test this hypothesis, we conduct the following experiment. We prompt the LLM to select between 1) a pair of candidate *next observations* given the current observation and action (probing knowledge of $p(o_{t+1}|a_t, o_{\leq t})$), or 2) a pair of candidate *actions* given the next observation and current observation (probing knowledge of $p(a_t|o_{t+1}, o_{\leq t})$). In each case, the pair contains the ground-truth and random sample. In this experiment, a $50\%$ accuracy corresponds to a random guess.

Results presented in Figure 2b show that the LLM performs relatively poorly on both of these tasks, indicating limited understanding of both the action space and the environment dynamics. This can potentially explain the limited performance of the LLM Policy approach on MiniWob-Hard, NetHack, and MetaWorld, while results on Wordle suggest that additional contributing factors are at play.

## 4 ANALYSIS OF AI FEEDBACK FOR RL

Our results so far suggest that, without additional fine-tuning, indirectly modeling policies by constructing reward functions through AI feedback is the most effective approach across the range of environments and modalities we studied. In this section, we examine how rewards shaped by this method can assist RL agents in addressing core decision-making challenges, such as credit assignment and exploration. Through this analysis, we also emphasize the ways in which reward misspecification can unintentionally arise and severely impair performance.

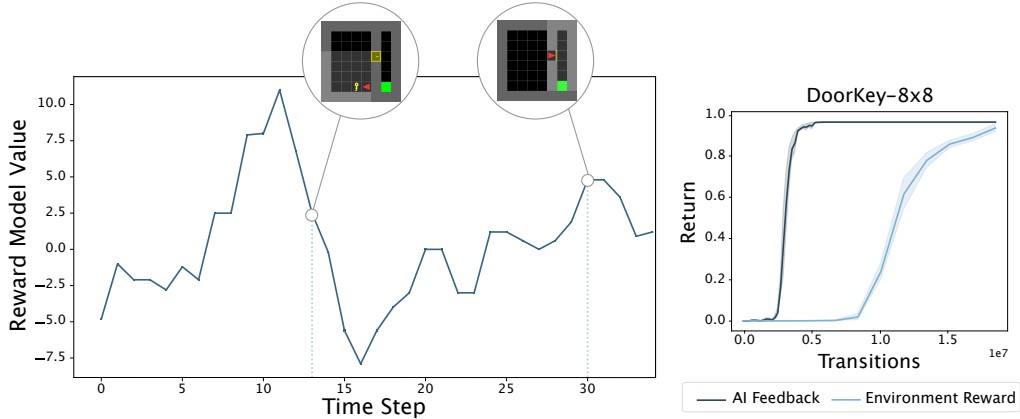

Figure 3: **Rewards learned through AI Feedback distribute rewards to key timesteps.** By doing so, the problem of credit assignment, or learning from delayed rewards, is significantly reduced. Such distribution effectively shortens the horizon over which the RL algorithm must propagate credit through its update rule.

### 4.1 CREDIT ASSIGNMENT

Setting aside performance of the direct policy method, we now turn our attention to why reward modeling with AI feedback is performing well. AI feedback-based rewards depend on the prompt

used to capture preferences. In the experiments conducted so far, these prompts were designed to elicit preferences by emphasizing states that contribute to task progress (see prompts Appendix A.2). Additionally, a key aspect of our methodology involved presenting the LLM with observations sampled randomly within trajectories. This enabled querying preference for any observation in the environment, rather limiting the focus to final states - a distinction also known as process-based and outcome-based reward models (Uesato et al., 2023; Lightman et al., 2023). What are the resulting characteristics of the reward model under such choices?

**Qualitative experiment** In Figure 3, we present the output of the AI feedback-based reward model over each timestep of an episode within a simple grid world environment. This task includes an agent, a key, a door, and a goal (Chevalier-Boisvert et al., 2023). We notice that this reward model naturally captures the fact that picking up the key, as well as opening the locked door, are important steps towards the goal. By propagating credit over such key moments in a trajectory, the LLM effectively shortens the horizon over which the RL algorithm must assign credit through temporal difference learning (Sutton & Barto, 2018). This is manifested in Figure 3 where the agent learning through AI feedback reaches a high success rate in a fraction of the timesteps required by a similar agent learning from the environment feedback (which in this case is sparse reward of +1 for reaching the goal). The dense reward resulting from LLM feedback can be seen as a form of reward redistribution (Arjona-Medina et al., 2018), which is an established method for improving credit assignment.

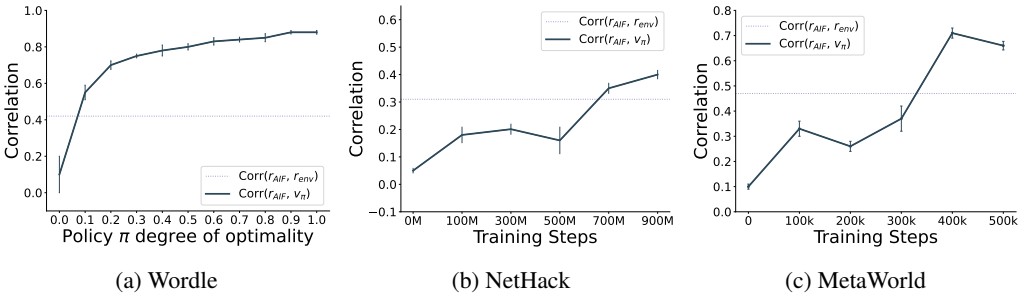

(a) Wordle  (b) NetHack  (c) MetaWorld

Figure 4: **LLM preferences correlate with value function preferences.** The correlation between Bradley-Terry models trained from frozen LLM state preferences and value function preferences increases as the online policy improves in 3 different domains.

**Quantitative experiment** In Figure 4, we present the correlation between the reward model derived from AI feedback and the value function of an RL agent across various levels of policy optimality. We observe that AI feedback generates reward functions with a stronger correlation to value functions obtained later in the training process compared to those from earlier stages. Additionally, this correlation is higher than that observed with the environment reward. In the Wordle game, we generate, in code, a near-optimal policy and estimate its value function using Monte Carlo. We then compare it to the LLM-derived reward function find an almost perfect correlation. These findings suggest that the reward models derived from AI feedback inherently encode aspects of high-quality value functions, which, when used as rewards for the RL agent, can substantially simplify the credit assignment process. In Appendix A.7, we provide additional insights from the lens of heuristic-guided reinforcement learning (Cheng et al., 2021).

## 4.2 EXPLORATION

In the previous section, we investigated how our standard prompting strategy can ease the problem of credit assignment in downstream RL tasks. This outcome stemmed from the specific preferences we requested from the LLM, that is, promoting task progress. However, to address different RL objectives, in particular the one of exploration, we may need to elicit alternative preferences.

Previously, Klissarov et al. (2024) employed AI feedback to design an effective reward function for an agent operating in the open-ended environment of NetHack. However, before applying this reward to the RL agent, the authors implemented the following transformation:

$$r(o_t) \propto r_{AIF}(o_t)/N(o_t)^\beta, \tag{4}$$

where $r_{AIF}$ is the reward model obtained from AI feedback, $N(o_t)$ denotes the number of times a particular observation $o_t$ was seen in an episode, and $\beta$ is a positive real-valued coefficient set to 3. The counting term was added to encourage exploration (Henaff et al., 2022), which is a key difficulty in NetHack. However, instantiating such a counting function proves difficult in many practical settings (Bellemare et al., 2016). Given the flexibility of natural language, can we alleviate the need for such a term and integrate the notion of exploration in the prompt itself?

In Figure 5, we demonstrate that this is indeed possible, leading to performance comparable when using count-based exploration by directly modifying the prompt used for preference elicitation. Specifically, when querying the LLM for preferences, we present it with a pair of sequences of observations (rather than a single observation) which provides crucial context. The prompt was also modified to steer the LLM towards avoiding low entropy sequences, i.e. sequences with repetitions (see Appendix A.2).

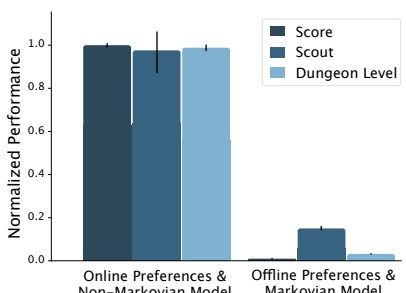

Figure 5: **By changing the prompt, LLMs can be steered to provide feedback that promotes exploration on NetHack.** Additionally, to avoid degenerate solutions, preferences should be elicited in an online fashion and the reward function be non-Markovian.

Our findings reveal two potential failure modes: the offline nature of the preference elicitation method and the assumption of a Markovian reward model. Previous research has demonstrated that online preference querying can outperform offline methods when aligning LLMs (Bai et al., 2022; Touvron et al., 2023). In our experiments, offline elicitation led to a performance collapse, likely due to frequent RL policy updates during online learning. Additionally, assuming a Markov reward model—where the current observation fully determines the reward—can lead to an equally poor performance, as complex tasks often require historical context beyond immediate observations (see Appendix A.8 for a full breakdown).

## 5 BEYOND ZERO-SHOT REWARD MODELING

So far, we have explored the ability of LLMs to model policies, directly and indirectly, without any fine-tuning. However, in many cases the prior knowledge encoded in LLM might not contain the necessary information to do so successfully. In such instances, fine-tuning becomes an effective method for incorporating task-specific knowledge into the model.

We consider the `sweep-into` task from MetaWorld, where AI feedback rewards lead to a success rate of only 15%. When measuring the perplexity score of the PaliGemma model on captions describing the pixel observations from the task, we obtain a value of 16.03. Both of these results indicate poor understanding and the necessity to adapt the model.

We therefore fine-tune PaliGemma on image-caption pairs annotated by GPT-4o and trained the model to predict the caption for a given image. Figure 6a shows significant gains in downstream RL performance after only a few fine-tuning epochs and as few as approximately 100 image-caption pairs. Moreover, Figure 6a shows how this procedure only marginally decreases performance of the LLM on the standard multi-modal reasoning benchmarks, such as POPE (Li et al., 2023b), GQA (Hudson & Manning, 2019), AI2D (Kembhavi et al., 2016) and MMMU (Yue et al., 2024). Surprisingly, performance on the AI2D benchmark *improves* as the number of RL-specific fine-tuning epochs increases.

We contrast these findings with Figure 6b, where we fine-tune PaliGemma with behaviour cloning on expert data on the same MetaWorld task. Similarly to RT-2 (Brohan et al., 2023), we overwrite the least frequent tokens with residual VQ-VAE codebooks (Szot et al., 2024). In this case, any significant increase of RL performance comes at the cost of catastrophically forgetting all previous knowledge. These results hint at an important trade-off: if preserving prior language reasoning knowledge is important, fine-tuning for AI feedback offers a viable approach. However, if maximizing downstream RL performance is the sole objective, directly fine-tuning for action selection can be more effective.

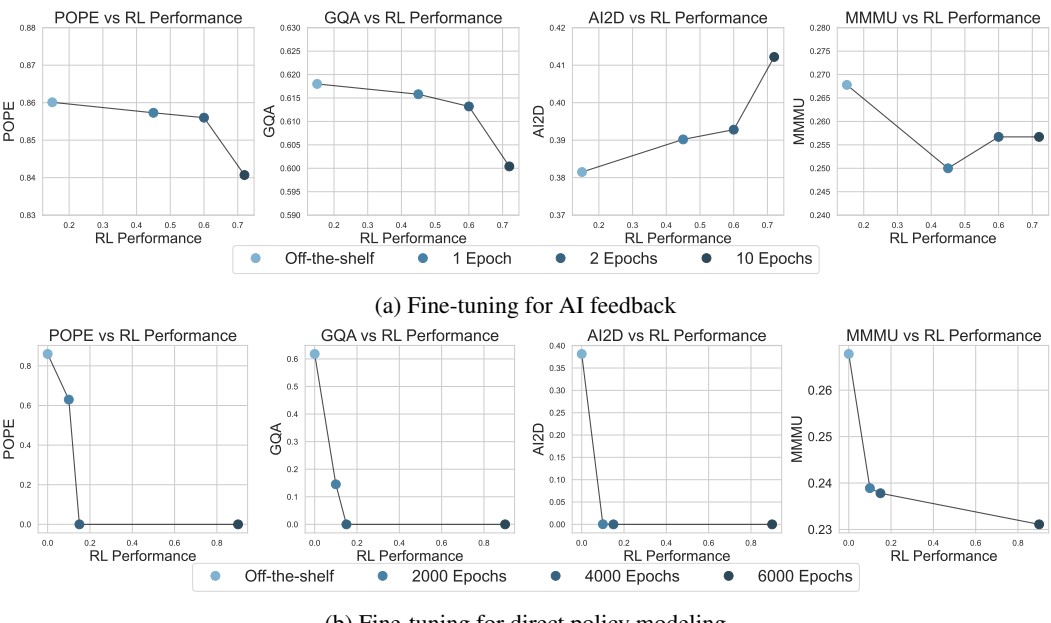

(a) Fine-tuning for AI feedback

(b) Fine-tuning for direct policy modeling

Figure 6: **Fine-tuning LLMs for AI feedback better preserves their prior knowledge.** LLMs fine-tuned for AI feedback in (a) retain a higher portion of their original language reasoning knowledge than those fine-tuned for direct action selection in (b).

## 6 RELATED WORKS

Large language models (LLMs) require additional adaptation for general-use language tasks (Christiano et al., 2017; Stiennon et al., 2020; Ouyang et al., 2022; Mialon et al., 2023). Without additional context and/or fine-tuning, LLMs can generate misleading, harmful, or even nonsensical answers to queries or conversations with humans (Bai et al., 2022). To modify their behavior, it is necessary to tune their prompts and/or fine-tune their outputs to ensure their output is desirable w.r.t. some set of linguistic tasks before deployment. This at least if not more true in embodied settings, where real-world actions can have physical consequences, and methodologies for modifying LLM behavior in embodied settings more-or-less align with efforts in the language space.

**Prompt tuning** Arguably the most common theme among techniques that modify LLM behavior in general is to change the prompt such that the distribution of LLM outputs better-fits a given desiderata on behavior. Prompt-engineering can greatly align or calibrate an LLM, pretrained or no, to desired beneficial behavior (Christiano et al., 2017; Glaese et al., 2022; Bai et al., 2022), or even expose harmful or other unexpected behaviors. Chain-of-thought (CoT, Wei et al., 2022) is an in-context method to either few-shot or zero-shot (Kojima et al., 2022) adjust an LLM's outputs to generate more correct responses to question-and-answering tasks. Further modifications to the prompt such as providing feedback from an environment (Yao et al., 2022), self-critique (Zelikman et al., 2022), or self-reflection (Shinn et al., 2023) can improve LLM performance in language as well as tasks that have an environment. The biggest promise of in-context-based methods in RL is that somewhere within the LLM's conditional distribution is the optimal policy for any given task (Brohan et al., 2023; Szot et al., 2023), an accurate world-explicit model (Lin et al., 2024), and/or a useful reward-model (Klissarov et al., 2024). For example, Sun et al. (2023) has shown strong results on text-based domains such as web agents by using LLMs as policies. Other approaches combine different quantities from the RL problem within the same algorithm, for example by providing LLM rewards/critiques as context to an LLM acting as a policy (Zhou et al., 2023; Liu et al., 2024c). LLMs have also been shown to be particularly good at devising high-level plans that will guide an agent acting in the environment (Nottingham et al., 2023).

**Querying model for feedback** Another hypothesis is that LLMs contain knowledge relevant to tasks, and this knowledge can be extracted (Xu et al., 2024) in a way to train a policy that has desirable

behavior (Huang et al., 2022). Kwon et al. (2023) first studies the possibility of querying an LLM to design reward functions. In their method, the LLM is given a task description, examples and an objective specified by a user, and is used to outputs text that is mapped to a binary value. Their results show that their approach is significantly more sample efficient compared to a supervised learning approach. The most successful reward modeling method in our paper is based on RL from AI Feedback (RLAIF Bai et al., 2022; Lee et al., 2023), a scalable method akin to but without the practical issues that come paired with RL from Human Feedback (RLHF Christiano et al., 2017), the goal of which is to fine-tune an existing LLM to be more specific, accurate, innocuous, etc. RLAIF trains a reward model on a dataset collected from an LLM's preferences given a dataset of language responses from an LLM and a given set of queries, and this reward model is used to train a policy using RL, for example using PPO. This process of extracting knowledge using preference data can also be directly used to train a policy without a reward model, as in Direct Preference Optimization (Rafailov et al., 2024). A related line of work studies the possibility of LLMs crafting their own rewards to improve their capabilities. Yuan et al. (2024) leverage DPO to improve an LLM's ability to for instruction following. Investigating this idea within the domain of sequential decision making is particularly promising.

## 7    DISCUSSION

In this paper, we explored two distinct approaches to leveraging LLMs for solving RL tasks: 1) directly, by modeling policies and 2) indirectly, by modeling rewards to be leveraged within a policy learning algorithm. Our results indicate that, without task-specific fine-tuning, current LLMs only show limited decision-making capabilities when directly generating actions. However, despite this limitation, LLMs are capable zero-shot reward modelers. In particular, when eliciting preferences to define rewards through the Bradley-Terry model, LLMs show strong performance across a wide range of domains presenting various challenges.

In cases where an LLM's prior knowledge is not enough to obtain useful reward functions, we also investigated fine-tuning with task-specific data to bridge this gap. Notably, fine-tuning to enhance reward modeling capabilities helps mitigate catastrophic forgetting, which is a crucial consideration for preserving the LLM's general-purpose abilities Maintaining these capabilities is essential for broad applicability to sequential decision-making tasks, including out-of-distribution tasks, and for supporting continued natural language interaction with users.

The reward modeling capabilities presented in this work offer potential solutions to challenges in RL. First and foremost, LLM-derived reward models alleviate the need for human-designed reward functions, which are often complex and costly to develop. Second, our empirical analysis reveals that AI-feedback based rewards produce dense functions which correlate positively with high-quality value functions. Such reward functions can significantly reduce the difficulty of assigning credit by redistributing rewards across different steps within a trajectory. Finally, distilling knowledge from LLMs into reward models opens new possibilities for applying RL in environments where simulators or symbolic features are unavailable—such as embodied AI agents interacting with humans. Additionally, when dealing with real-world scenarios, we may be concerned by the speed at which a model may react to its environment. Direct policy modeling implies that the LLM is executed in the environment, which might limit the frequency with which it can queried. For example, the report from Black et al. (2024) emphasizes the necessity of using special techniques such as action chunking to achieve a 50 Hz control frequency. In contrast, indirect policy modeling can distill an LLM's knowledge into a smaller neural network, which would be queried much faster.

Some notable limitations and caveats exist. It is possible that by using LLM feedback to design reward functions, we may obtain contradictory and inconsistent preferences, simply by virtue of LLMs having an imperfect understanding of the task at hand. In fact, we believe this is the main reason why, without fine-tuning, the PaliGemma model produces some unsuccessful RL policies on MetaWorld, as seen in Figure 6. If a particular task requires detecting subtle, incremental progress, it is entirely possible that the LLM might miss key milestones. Additionally, interacting with LLMs through natural language requires experimenting with various prompting techniques and specifications. However, this flexibility also enables the shaping of reward functions to incorporate valuable strategies (Knox et al., 2013), such as promoting exploration, which can further enhance the performance of RL agents.

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

# A  APPENDIX

## A.1  ENVIRONMENT DETAILS

In our experiments, we investigate tasks from four different domains: MiniWob (Liu et al., 2018), NetHack (Küttler et al., 2020), and Wordle (Lokshtanov & Subercaseaux, 2022), and MetaWorld (Yu et al., 2019). The observation space for all these environments is text, except fro MetaWorld which consists of RGB pixels.

In the MiniWob domain, we sample the subset of the five tasks on which state-of-the-art results are low. Specifically, we carry experiments on: `click-tab-2-hard`, `click-checkboxes-soft`, `count-shape`, `tic-tac-toe` and `use-autocomplete`. To learn RL policies from LLM-based rewards, we leverage the experimental setup of Shaw et al. (2023). In NetHack, we use the same environment and the same algorithmic setup as in Klissarov et al. (2024). In Wordle, we build on the code made available by Snell et al. (2022) and use their proposed subset of 200 words from the official list of the game. Finally, in MetaWorld we study the same subset of environments presented in (Wang et al., 2024) consisting of `drawer-open-v2`, `soccer-v2` and `sweep-into-v2`. Across all experiments where RL policies are learned, we use the original hyperparameter values defined in the respective experimental setups we are building upon.

## A.2  DETAILS ON INDIRECT POLICY MODELING THROUGH LLM-BASED REWARDS

We use the following prompt templates to query the agent for AI feedback, Scalar Reward and Reward as Code across various environments. For the Embedding-based approach, we use calculate the cosine similarity between the representation, provided by a BERT (Devlin et al., 2019) sentence encoder (specifically the same `paraphrase-MiniLM-L3-v2` model) when environments are text-based, and otherwise we use the CLIP encoder (Radford et al., 2021). The similarity is measured between the current observation and the same goal description contained in the each of the following prompts given for the other baselines.

---

**MiniWob Prompt For Reward Modeling with AI feedback**

```
I will present you with two HTML descriptions from a web interaction
environment.

{task_description}
Write an analysis describing the semantics of each description
strictly using information from the descriptions.
Provide a comparative analysis based on first principles.
Finally, express a preference based on which description is the
most likely to make some progress towards the goal, writing either
("best_description":  1), ("best_description":  2).
You could also say ("best_description":  None).

html_description_1:  {description_1}

html_description_2:  {description_2}
```

---

Prompt 1

In Figure 7, we verify the importance of the chain-of-thought prompting used for our AI feedback baseline. Results show no statistical difference in performance compared to using chain-of-thought. To understand this result, we can refer to Sprague et al. (2024) which show that across 14 LLMs, chain-of-thought only significantly helps for mathematical and symbolic problems. We believe the way LLMs help the tasks studied in this paper (i.e., for generating feedback on trajectories of experience) can be characterized as being part of the Commonsense and Knowledge categories from Sprague et al. (2024), in which chain-of-thought does not improve performance.

**Wordle Prompt For Reward Modeling with AI feedback**

```
I will present you with two short gameplay descriptions of Wordle.

First, tell me about your knowledge of Wordle.

Mention the goal of Wordle.  Use the following information for Wordle
states:  black means that the provided letter is not present anywhere
in the hidden word.  yellow means that the provided letter is present
somewhere in the hidden word, but not at the correct position.  green
means that the provided letter is present in the hidden word exactly
at the correct position.

Then, write an analysis describing the semantics of each description
strictly using information from the descriptions (which may be empty)
and your knowledge of Wordle.
Provide a comparative analysis based on first principles.
Finally, express a preference based on which description is the
most likely to make some progress towards the goal, writing either
("best_description":  1), ("best_description":  2).
You could also say ("best_description":  None).

description_1:  {description_1}

description_2:  {description_2}
```

Prompt 2

**NetHack Prompt For Reward Modeling with AI feedback**

```
I will present you with two short gameplay descriptions of Nethack.

First, tell me about your knowledge of NetHack.

Mention the goal of NetHack.  Prefer agents that maximize the score
in the game, for instance by killing monsters, collecting gold or
going down the stairs in the dungeon.

Then, write an analysis describing the semantics of each description
strictly using information from the descriptions (which may be empty)
and your knowledge of NetHack.
Provide a comparative analysis based on first principles.
Finally, express a preference based on which description is the
most likely to make some progress towards the goal, writing either
("best_description":  1), ("best_description":  2).
You could also say ("best_description":  None).

description_1:  {description_1}

description_2:  {description_2}
```

Prompt 3

---

**NetHack Prompt For Online Reward Modeling with AI feedback**

```
I will present you with two short gameplay descriptions of Nethack.

First, tell me about your knowledge of NetHack.

Mention the goal of NetHack.  Prefer agents that maximize the score
in the game, for instance by killing monsters, collecting gold or
going down the stairs in the dungeon.

Then, write an analysis describing the semantics of each description
strictly using information from the descriptions (which may be empty)
and your knowledge of NetHack.
Provide a comparative analysis based on first principles.
Finally, express a preference based on which description is the
most likely to make some progress towards the goal, writing either
("best_description":  1), ("best_description":  2).
You could also say ("best_description":  None).

description_1:  {description_1}

description_2:  {description_2}
```

---

Prompt 4

---

**MetaWorld Prompt For Reward Modeling with AI feedback**

```
Does the image satisfy {current_task}?
image_1:  {image_1}
{llm_response}

Does the image satisfy {current_task}?
image_2:  {image_2}
{llm_response}
```

---

Prompt 5

## A.3  Details on Direct Policy Modeling

We present the exact prompts used to query GPT-4o for each of the domains we have considered. These are presented through Prompt 13, 15, 14 and 16.

Additionally, in Figure 8, we ablate the prompting techniques used in our direct policy modeling approach. Results show that a combination of all prompting techniques presented in Section 2.1 works best.

## A.4  Additional Indirect Policy Modeling Methods

There are a number of other prompting methods for extracting information or *knowledge* from an LLM that may be relevant to solving RL tasks.

- **Direct State Generation**. The model generates tokens that will represent next states (or other-future-time states). This is similar to world modeling. The next state prediction can be conditioned on an action, or marginalized over a policy distribution.

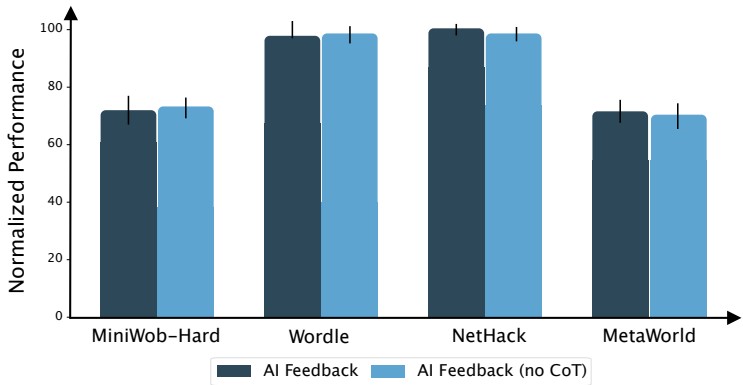

Figure 7: Ablation on the usefulness of chain-of-thought in the AI Feedback baseline.

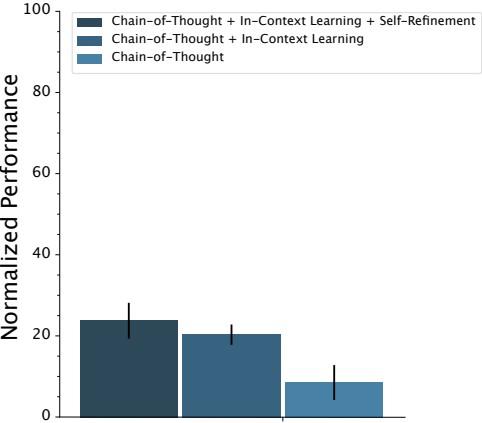

Figure 8: Ablation on the set of prompting techniques used for direct policy modeling. The reported performance is averaged over all domains and tasks.

---

**MiniWob Prompt For Reward Modeling with Scalar Reward**

```
I will present you with an HTML descriptions from a web interaction
environment.

{task_description}

Write an analysis describing the semantics of the description
strictly using information from the description.

Finally, output a scalar value between 0 and 5, with higher values
correlation with progress towards the goal.

html_description:  {description}
```

---

Prompt 6

---

**Wordle Prompt For Reward Modeling with Scalar Reward**

```
I will present you with a gameplay description of Wordle.

First, tell me about your knowledge of Wordle.

Mention the goal of Wordle.  Use the following information for Wordle
states:  black means that the provided letter is not present anywhere
in the hidden word.  yellow means that the provided letter is present
somewhere in the hidden word, but not at the correct position.  green
means that the provided letter is present in the hidden word exactly
at the correct position.

Write an analysis describing the semantics of the description
strictly using information from the description.
Finally, output a scalar value between 0 and 5, with higher values
correlation with progress towards the goal.
description:  {description}
```

---

Prompt 7

- **High-level plans**. Ask the LLM to generate a high-level plan. This can be similar to direct state generation, but potentially relies on some subset of features (for examples the inventory in MineCraft) and is typically temporally extended.
- **Action Preference**. Ask the LLM to select, among two choices, the most likely action given previous and future observations.
- **State Preference**. Ask the LLM to select, among two choices, the most likely next state or observation conditioned on prior history and/or actions.

Many of the above could theoretically be used to construct a policy, yet a full implementation is out of scope from this paper due to the lack of available code-bases to build upon and we do not seek to build new algorithms from scratch. For instance, generating a high-level plan can be constructed in various ways: ranging from in-context learning to hierarchical reinforcement learning agents, with many variables to be decided upon. For example, what is the space over which plans are established? Is it the full observation space, a subset of it (e.g., in Minecraft, should it be pixel space or the inventory)? What action space should be considered (e.g., in web agents tasks should it be low level mouse click and drag, or human-crafted high levels as we see in many baselines?)? Additionally, the

---

**NetHack Prompt For Reward Modeling with Scalar Reward**

```
I will present you with a gameplay description of Nethack.

First, tell me about your knowledge of NetHack.

Mention the goal of NetHack.  Prefer agents that maximize the score
in the game, for instance by killing monsters, collecting gold or
going down the stairs in the dungeon.

Write an analysis describing the semantics of the description
strictly using information from the description.
Finally, output a scalar value between 0 and 5, with higher values
correlation with progress towards the goal.
description:  {description}
```

Prompt 8

---

**MetaWorld Prompt For Reward Modeling with Scalar Reward**

```
From 0 to 5, how much does the image achieve{current_task}?
image:  {image}
```

Prompt 9

---

**MiniWob Prompt For Reward Modeling with Reward as Code**

```
I will present you with HTML descriptions from a web interaction
environment.

{task_description}

Write an analysis describing the semantics of the descriptions
strictly using information from the descriptions.

Finally, write a code that, when executed, will help make progress
towards the goal.

html_descriptions:  {descriptions}
```

Prompt 10

---

choice of these variables might not easily generalize across many environments, which is why we decided not to focus on this direction. We do believe that this is a particularly important direction for future work.

However, in Figure 2b we perform investigations into the capabilities of LLMs to perform Action Preference and State Preference. The results show that current LLMs struggle to achieve strong performance on any of these tasks. Additionally, in Table 1, we report the accuracy with which LLMs directly predicts the next observation (Direct State Generation), providing a probe into their direct world modeling capabilities. Results show limited performance, except on MiniWob-Hard tasks, which are fully observable and encode deterministic transitions.

---

**Wordle Prompt For Reward Modeling with Reward as Code**

```
I will present you with gameplay descriptions of Wordle.

First, tell me about your knowledge of Wordle.

Mention the goal of Wordle.  Use the following information for Wordle
states:  black means that the provided letter is not present anywhere
in the hidden word.  yellow means that the provided letter is present
somewhere in the hidden word, but not at the correct position.  green
means that the provided letter is present in the hidden word exactly
at the correct position.

Write an analysis describing the semantics of the descriptions
strictly using information from the description.
Finally, write a code that, when executed, will help make progress
towards the goal.
descriptions:  {descriptions}
```

Prompt 11

---

**NetHack Prompt For Reward Modeling with Reward as Code**

```
I will present you with gameplay descriptions of Nethack.

First, tell me about your knowledge of NetHack.

Mention the goal of NetHack.  Prefer agents that maximize the score
in the game, for instance by killing monsters, collecting gold or
going down the stairs in the dungeon.

Write an analysis describing the semantics of the descriptions
strictly using information from the descriptions.
Finally, write a code that, when executed, will help make progress
towards the goal.
descriptions:  {descriptions}
```

Prompt 12

## A.5 ABLATING REWARD AS CODE

In Table 2, we ablate the performance of the Reward as Code baseline across LLMs, observation spaces and additional assumptions. For pixel observations, we follow the methodology laid out in

|              | Accuracy        |
| ------------ | --------------- |
| MiniWob-Hard | $65 \pm 11.4\%$ |
| Wordle       | $28 \pm 8.3\%$  |
| NetHack      | $0.0 \pm 0.0\%$ |
| MetaWorld    | N/A             |

Table 1: **LLMs struggle to predict the next observation.** We show the decreasing accuracy of the LLM to predict the next observation with increasing task complexity. LLMs are unable to generate pixel observations, which are used in MetaWorld.

---

**MiniWob-Hard Prompt For Direct Policy Modeling**

We have an autonomous computer control agent that can perform atomic
instructions specified by natural language to control computers.
There are two types of instructions it can execute.

First, given the instruction that matches the regular expression,
"^type.{1,}$"
Second, given the instruction that matches the regular expression,
"^clickxpath\s.{1,}$" it can click an HTML element with an xpath that
is visible on the webpage.  The target of this instruction should be
a valid xpath.

Below is the HTML code of the webpage where the agent should solve
a task.

{html_observation}

Examples:
task:  {example_task}
plan:  {example_plan}

Current task:  Enter an item that starts with Äntiänd ends with ̈da.̇
Think step-by-step before answering, what is the current plan?
{llm_plan}

===============
Repeat N times:

Find problems with this plan for the given task compared to the
example plans.

{llm_criticism}

Based on this, what is the plan for the agent to complete the task?

Below is the HTML code of the webpage where the agent should solve
a task.
{html_observation}

Current task:  Enter an item that starts with Äntiänd ends with ̈da.̇
Think step-by-step before answering, what is the current plan?
{llm_plan}
===============

---

Prompt 13

(Venuto et al., 2024), whereas for proprioceptive observations we follow the one from (Yu et al., 2023). Both methods heavily depend on access to a state-of-the-art, closed-source model to achieve performance comparable to that of AI Feedback, which uses the smaller, open-source model of Paligemma (Beyer et al., 2024).  Additionally, each method requires expert demonstrations or specialized domain knowledge to guide the reward design process. While these assumptions may be viable in certain situations, such as in a controlled simulation environment, they can present significant practical challenges in more general contexts. In contrast, AI Feedback operates by simply comparing observations and reasoning using a chain-of-thought approach.

---

**Wordle Prompt for Direct Policy Modeling**

```
Let's play a game of Wordle.  You will have to guess the words and I
will give you the colors.

Use the following information for Wordle colors:
black means that the provided letter is not present anywhere in the
hidden word.
yellow means that the provided letter is present somewhere in the
hidden word, but not at the correct position.
green means that the provided letter is present in the hidden word
exactly at the correct position.

You can choose among this list of words:  {list_of_words}

Here are examples trajectories, containing past observations and
actions, together with an appropriate action.

Example 1:
Trajectory:  {example_trajectory}
Action:  {example_action}

Example 2:
Trajectory:  {example_trajectory}
Action:  {example_action}

Current trajectory:  {trajectory_so_far}
Think step-by-step before answering, what should be the current
action?  {llm_action}

===============
Repeat N times:

Find problems with this action for the given task compared to the
example actions.

{llm_criticism}

Based on this, what is the action for the agent to make progress on
the task?

Current trajectory:  {trajectory_so_far}
Think step-by-step before answering, what should be the current
action?  {llm_action}
===============
```

---

Prompt 14

## A.6 Learning from Environment Rewards

In Figure 9, we compare the performance of an RL agent trained using a reward function derived from AI feedback with that of an agent trained on human-designed rewards across different environments. We observe that AI feedback achieves comparable results, with an average score of $89.93$ versus $86.3$ for the human-designed reward. The objective of this experiment is not to argue that LLM-based rewards consistently outperform human-crafted ones—since expert human knowledge can always be encoded into a reward function—but rather to contextualize the performance of LLM-based rewards. Notice that for MetaWorld we report the performance after fine-tuning the LLM as described in Section 5.

---

**NetHack Prompt for Direct Policy Modeling**

```
Let's play the game of NetHack.

First, tell me about your knowledge of NetHack.  Mention the goal
of NetHack.

Prefer maximizing the score in the game, for instance by killing
monsters, collecting gold or going down the stairs in the dungeon.

Here are examples sub-trajectories, containing past observations and
actions, together with an appropriate action.

Example 1:
sub-Trajectory:  {example_sub-trajectory}
Action:  {example_action}

Example 2:
sub-Trajectory:  {example_sub-trajectory}
Action:  {example_action}

Current sub-trajectory:  {sub-trajectory_so_far}
Think step-by-step before answering, what should be the current
action?  {llm_action}

==============
Repeat N times:

Find problems with this action for the given task compared to the
example actions.

{llm_criticism}

Based on this, what is the action for the agent to make progress on
the task?

Here is the current sub-trajectory, containing past observations and
actions:  {sub-trajectory_so_far}
Think step-by-step before answering, what should be the current
action?  {llm_action}
==============
```

---

Prompt 15

## A.7 AI FEEDBACK AND HEURISTIC FUNCTIONS

While prior works have shown that rewards can be extracted from a language model (Kwon et al., 2023; Brooks et al., 2024; Klissarov et al., 2024), it can be more generally thought of as encoding a heuristic function $h$. The function $h$ contains high-level, multi-step information about the MDP $M$. To extract it, one can solve the re-shaped MDP $\tilde{M}$ with $\tilde{r}(s_t, a_t) = r(s_t, a_t) + (1 - \lambda)\gamma\mathbb{E}_{s_{t+1}|s_t,a_t}[h(s_{t+1})]$ and $\tilde{\gamma} = \lambda\gamma$ where $\lambda \in [0, 1]$ Cheng et al. (2021). Solving $\tilde{M}$ yields a policy $\pi^*$ that is also optimal in $M$ - its value function's bias can be shown to converge to $V^*$ in $M$ as a function of $||h - V^*||_\infty$.

Specifically, assume access to an initial dataset $\mathcal{D}_0$, from which a heuristic $h$ can be computed. In the reshaped MDP $\tilde{M}$, one can learn a new policy $\pi$ which optimizes $\tilde{r}$ with $\lambda \in [0, 1]$. Equation (5) shows the performance difference lemma Kakade & Langford (2002) as a function of true and reshaped MDP quantities:

---

**MetaWorld Prompt for Direct Policy Modeling**

```
You are controlling a robot for the following task:

{meta_world_task}

Here are examples sub-trajectories, containing past observations and
actions, together with an appropriate action.

Example 1:
sub-Trajectory:  {example_sub-trajectory}
Action:  {example_action}

Example 2:
sub-Trajectory:  {example_sub-trajectory}
Action:  {example_action}

Current sub-trajectory:  {sub-trajectory_so_far}
Think step-by-step before answering, what should be the current
action?  {llm_action}

==============
Repeat N times:

Find problems with this action for the given task compared to the
example actions.

{llm_criticism}

Based on this, what is the action for the agent to make progress on
the task?

Here is the current sub-trajectory, containing past observations and
actions:  {sub-trajectory_so_far}
Think step-by-step before answering, what should be the current
action?  {llm_action}
==============
```

---

Prompt 16

$$\mathcal{L}(\pi, h) = \mathbb{E}_{\mathcal{D}_0}[V^*(s) - V^\pi(s)]$$
$$= c_1 \mathbb{E}_{\mathcal{D}_0}\left[\tilde{V}^*(s) - \tilde{V}^\pi(s)\right] + c_2 \mathbb{E}_{\mathcal{D}^\pi}\left[\tilde{V}^*(s) - \tilde{V}^\pi(s)\right] + c_3 \mathbb{E}_{\mathcal{D}^\pi}\left[h(s') - \tilde{V}^*(s')\right], \quad (5)$$

where $c_1, c_2, c_3$ are non-negative constants. Minimizing $\mathcal{L}(\pi, h)$ with respect to $\pi$ and $h$ can be achieved by minimizing each individual term. In particular, the red term suggests that the heuristic $h$ has to be updated on data from $\mathcal{D}^\pi$ in order to not become "stale". This points out a shortcoming of existing LLM-as-critic algorithms, which sometimes fix $h$ after distilling the language model knowledge into it Klissarov et al. (2024)

These theoretical findings suggest, in particular, that heuristic $h$ (in our case, the Bradley-Terry preference model), has to be updated with on-policy samples, similarly to empirical results from Figure 5.

A.8    ADDITIONAL CONSIDERATIONS FOR PREFERENCE-BASED REWARD MODELING

In Figure 10, we present the properties that were important to obtain effective exploration on NetHack, without the counting term shown in Equation 4.

| Reward as Code - RGB Observations | |
|---|---|
| GPT-4o | |
|     w/o expert demonstration | $0\% \pm 1\%$ |
|     with expert demonstration | $79\% \pm 7\%$ |
| **Reward as Code - Proprioceptive Observations** | |
| Llama 3 70B | |
|     w/o background functional knowledge | $0\% \pm 1\%$ |
|     with background functional knowledge | $10\% \pm 3\%$ |
| GPT-4o | |
|     w/o background functional knowledge | $5\% \pm 3\%$ |
|     with background functional knowledge | $76\% \pm 6\%$ |
| **AI Feedback - RGB Observations** | |
| PaliGemma | $72\% \pm 8\%$ |

Table 2: **AI Feedback performs on par with Reward as Code, without proprioceptive observations or expert demonstrations.** To match AI Feedback performance on Metaworld, Reward as Code requires GPT-4o level knowledge, augmented with either in-context expert demonstrations or proprioceptive observations.

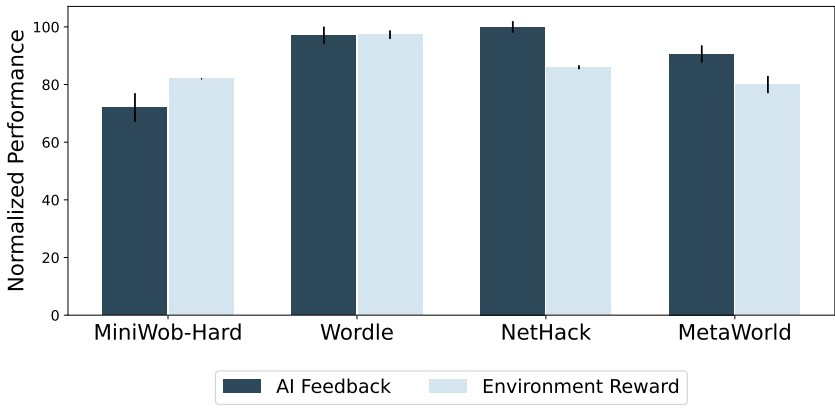

Figure 9: Comparison between the best performing LLM-based reward (AI Feedback) and human designed rewards for each domain.

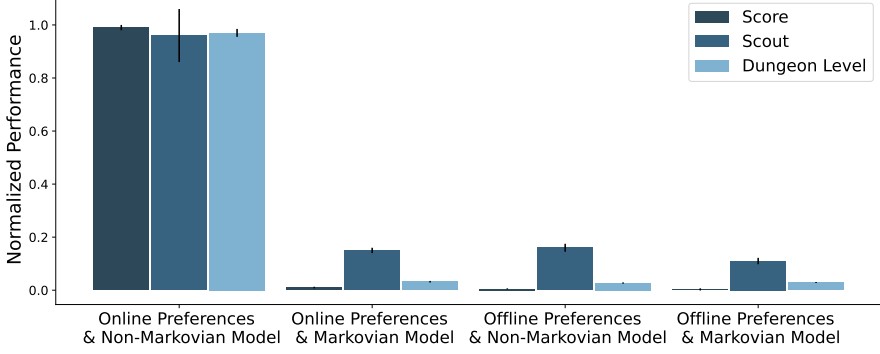

Figure 10: Successful exploration on Nethack depends on both online preference elicitation and a non-Markovian reward function.

### A.9    IN-CONTEXT LEARNING FOR REWARD MODELING

In Figure 11, we present a variation on the Wordle game where the color code has been altered, which we refer to as Eldrow (reverse Wordle). Under this transformation, the off-the-shelf model provides feedback that correlates very poorly with the optimal value function. When we measure the perplexity of the LLM on a natural language description of the new rule set of Eldrow (see Appendix A) we obtain a value of 6.97 which is higher than the one measured on the standard rule set of Wordle, with a value of 5.06. Given that the difference in values is not very large, we leverage the simplest way for adapting the LLM: through in-context learning. As shown in Figure 11b, by providing hints in the prompt about the new rule set, the LLM adapts its preferences and generates a Bradley-Terry model that recovers the correlation values we witnessed in 4.

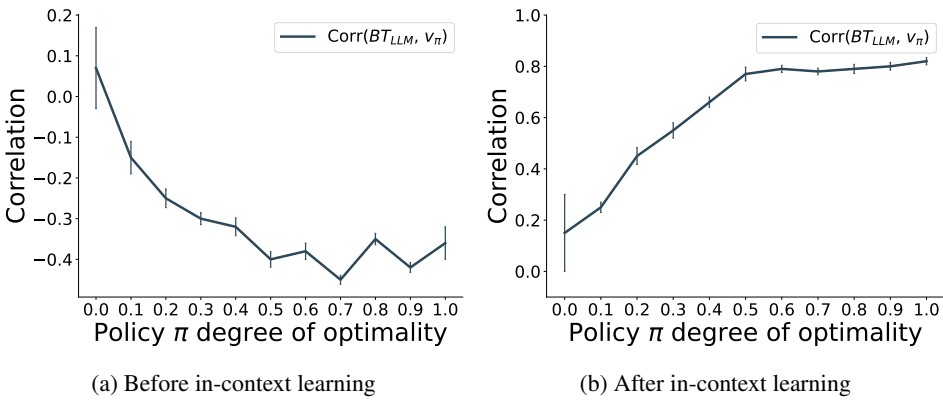

(a) Before in-context learning          (b) After in-context learning

Figure 11: **AI feedback can be adapted to novel settings through in-context learning.** While the original LLM does poorly on Eldrow due to out-of-distribution, it manages to correct its feedback the task using in-context hints.

### A.10    LLMS AS NOVELTY DETECTORS

We hypothesize that LLMs with long contexts can effectively act as novelty detectors. Within the scope of RL problems, this implies the ability to tell, for example, whether a sub-trajectory is contained in the replay buffer.

To test this, we query `Gemini-1.5 Pro` (Team et al., 2023) with a context video containing 500 frames of an agent exploring the bottom-left room (Figure 12-left) and a single frame sampled uniformly at random from a query episode which covers in the top-right room, center and bottom of the maze (Figure 12-middle). We ask the LLM to identify novel query states, i.e. states which are not seen in the context episode. We then train a direct predictor (3-layer MLP) to estimate the probability of any state on the grid to be novel with respect to the context (Figure 12-right). The language model correctly identifies the top-right portion of the trajectory to be novel, knowledge which could then be used to construct an intrinsic reward function.

### A.11    INVESTIGATING A LARGER SET OF LLMS

In Figure 13, we investigate a larger set of LLMs for the LLM Policy method. We run experiments using Claude 3.5 Sonnet and Gemini Pro 1.5 across all environments. Results indicate that the performance of these frontiers models is generally the same, mirroring recent results from Paglieri et al. (2024). In Figure 14, we extend the of LLM studied for the AI Feedback method, which is the best performing method from the reward modeling approaches. We investigate the Qwen 2.5 model on MiniWob-Hard, Wordle and NetHack (as they are text-based domains), and the Llama 3.2-V model on MetaWorld (pixel-based). Once again, the results show little difference in performance across most models, except for the comparison on the MetaWorld domain where the Llama 3.2-V model performs significantly better than PaliGemma.

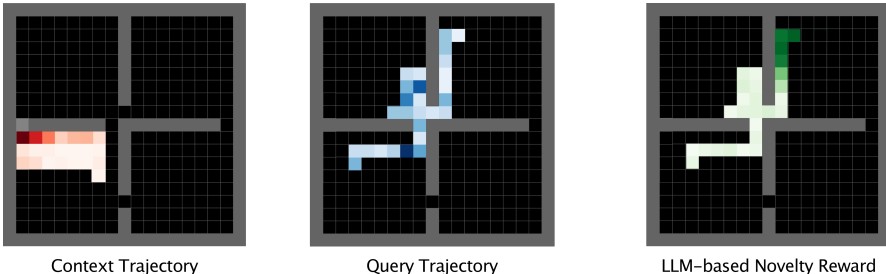

Context Trajectory     Query Trajectory     LLM–based Novelty Reward

Figure 12: **LLMs can capture observation novelty.** Given the context trajectory (red), and a single observation sampled uniformly at random from the query trajectory (blue), the LLM correctly identifies novel states that are seen in the query but not in the context (green).

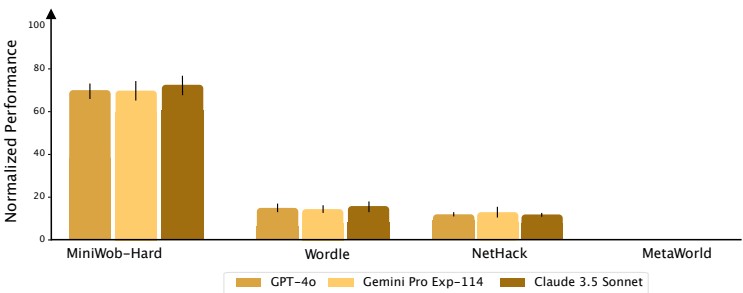

Figure 13: Comparison between different frontier closed-source models as used within the LLM Policy method.

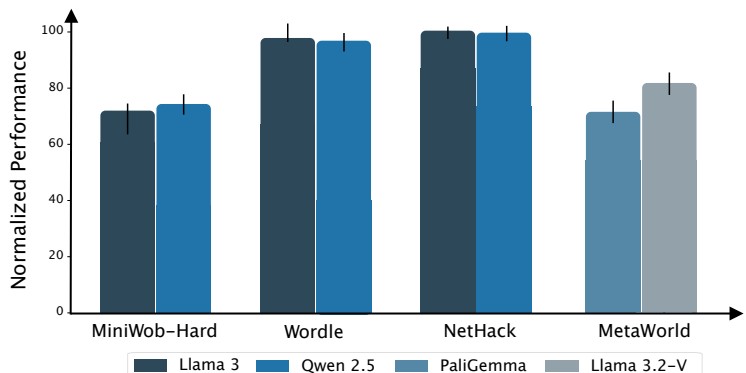

Figure 14: Comparison between the Llama 3 model and Qwen 2.5 on the three text-based domains (MiniWob-Hard, Wordle and NetHack) as well as a comparison between the PaliGemma model and Llama 3.2-V on the pixel-based domain (MetaWorld). We only notice significant difference in performance between PaliGemma and Llama 3.2-V, which also present quite different architectures and pre-training regimes.

## A.12   REWARD AS CONTEXT FOR LLM POLICY

In Figure 15, we verify whether providing the LLM-based reward as additional context to the LLM Policy can improve its performance.

## A.13   LLM FINE-TUNING EXPERIMENTS

In this section, we investigate how we could fine-tune an LLM for direct policy modeling in the environments we have studied. During this process, we control the amount of samples seen during

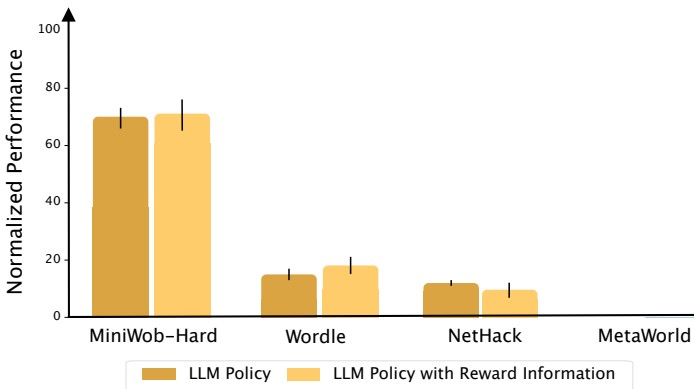

Figure 15: Comparison between the LLM Policy described in Section 2.2 and a variation, LLM Policy with Reward Information, in which we provide the LLM outputting actions in the environment with the reward defined from the AI Feedback method.

| Environment Samples | Direct Policy (only RL) | Direct Policy (only SFT) | Direct Policy (RL after SFT) | AI Feedback |
|---|---|---|---|---|
| | | Success Rate | | |
| 0 | 0.00 | 0.00 | 0.92 | 0.00 |
| 128k | 0.01 | 0.10 | 0.80 | 0.22 |
| 256k | 0.05 | 0.13 | 0.63 | 0.48 |
| 1M | 0.09 | 0.87 | 0.65 | 0.72 |
| | | Floating Point Operations (FLOPS) | | |
| 0 | 0 | 0 | $3.65 \times 10^{17}$ | $6.9 \times 10^{12}$ |
| 128k | $7.36 \times 10^{16}$ | $4.38 \times 10^{17}$ | $4.38 \times 10^{17}$ | $4.09 \times 10^{14}$ |
| 256k | $1.46 \times 10^{17}$ | $5.11 \times 10^{17}$ | $5.11 \times 10^{17}$ | $8.109 \times 10^{14}$ |
| 1M | $7.12 \times 10^{17}$ | $1.09 \times 10^{18}$ | $1.09 \times 10^{18}$ | $3.145 \times 10^{15}$ |

Table 3: FLOPS for Different Policies and Environment Samples.

optimization and the amount of FLOPS used to fine-tune the LLM. Specifically, we fine-tune the direct policy method under three settings: only with online RL, only with SFT, and online RL training on an SFT-pretrained model. We compare these to the performance of the AI feedback method, which has shown to be the best performing reward modeling method. For AI feedback, we calculate the FLOPS used by the RL algorithm, as well as any computation spent on the LLM for inference or fine-tuning.

Results in Table 3 indicates that indirect policy modeling achieves comparable performance to direct policy modeling trained with SFT at a fraction of the computational cost, i.e. two orders of magnitude lower. Policy modeling achieves a higher success rate with 1 million training samples, but is otherwise significantly less efficient than indirect policy modeling. Importantly, direct policy modeling with SFT relies on a large expert dataset to achieve good performance. On the other hand, direct policy modeling trained only with RL shows poor performance with 1M samples and high sensitivity to hyperparameters. This is in line with Anonymous (2024) reporting low sample efficiency when training LLMs with online RL, without SFT, for sequential decision-making.

