# OpenReview forum: "On the Modeling Capabilities of Large Language Models for Sequential Decision Making"
_ICLR.cc/2025/Conference — ICLR 2025 Poster_

### Official Review · Reviewer_9H9A · 2024-10-31

**Soundness:** 3
**Presentation:** 3
**Contribution:** 2
**Rating:** 6
**Confidence:** 4

**Summary:**

The paper investigates how LLMs can be leveraged for RL, comparing direct policy generation versus indirect reward modeling approaches across diverse domains. The authors find that using LLMs to generate reward models, particularly through AI feedback preferences, yields better and more consistent performance compared to direct policy generation. They also explore fine-tuning approaches and analyze how LLM-based rewards can help address core RL challenges.

**Strengths:**

- Comprehensive empirical evaluation across diverse domains.
- Interesting exploration of fine-tuning trade-offs between direct and indirect approaches.
- The paper is well written and easy to follow.

**Weaknesses:**

See the questions below.

**Questions:**

- The proposed method relies on consistent preference feedback from LLMs to construct reward signals, but doesn't adequately address cases of preference uncertainty. Consider a scenario where the LLM assigns similar preference probabilities to different states, or worse, provides inconsistent rankings when the same state pair is presented multiple times with slightly different prompts.
	- So, what happens when the LLM expresses low confidence or contradictory preferences?
	- Could inconsistent preferences lead to unstable reward signals that harm policy learning?
- The paper demonstrates the method primarily on environments where progress is relatively obvious (e.g., clear Wordle feedback, discrete game states in NetHack). However, many real-world tasks involve subtle, continuous progress where improvements may be hard to detect from observations. I have several concerns regard this problem:
	- How sensitive is the LLM's preference detection to small state changes?
	- Could the method miss important incremental progress by only detecting large, obvious changes?
- Sec 4.1 shows some evidence that AI feedback can help with credit assignment, the underlying mechanism isn't clear. In complex tasks, progress often results from a sequence of coordinated actions rather than single decisions. Can the method distinguish between critical and auxiliary actions when both contribute to the final outcome?

---

> ### Author Response · Authors · 2024-11-22
>
> >The proposed method relies on consistent preference feedback from LLMs to construct reward signals, but doesn't adequately address cases of preference uncertainty. Consider a scenario where the LLM assigns similar preference probabilities to different states, or worse, provides inconsistent rankings when the same state pair is presented multiple times with slightly different prompts.
> -So, what happens when the LLM expresses low confidence or contradictory preferences?
> -Could inconsistent preferences lead to unstable reward signals that harm policy learning?
>
> It is possible that with current LLMs we will obtain contradictory and inconsistent preferences, simply by virtue of LLMs being imperfect. In fact, we believe this is the main reason why, without fine-tuning, the PaliGemma model produces some unsuccesful RL policies on MetaWorld. To accurately measure the impact of the feedback the LLM provides, we want to re-iterate that the RL agent learns *only* through the reward model distilled from these LLM preferences, without access to any other reward. Therefore, inconsistent or contradictory preferences can lead to a collapse in behaviour.
>
>
> From this poin of view, our results in Section 5 become particuarly important. These results show that, with only a handful of transitions for LLM fine-tuning, it is possible to significantly improve the quality of the LLM feedback, and, consequently, the performance of the RL agent. We can witness this improvement through the following qualitative figures [before](https://imgur.com/a/pIEmkrq) and [after](https://imgur.com/a/QJkZDRK) fine-tuning. We notice that the LLM preferences, after fine-tuning, have become more consistent with the task at hand.
>
> We will be including these additional points of dicussion raised by the reviewer in the revised version of the paper.
>
> We also want to highlight our results in Section 4.2, which illustrate a phenomenon, separate from the quality of the LLM, that can harm policy learning. In this case, while the LLM provides useful feedback, its knowledge is distilled into a Markovian Bradley-Terry reward model, which is not able to correctly represent the LLM preferences. These results show how the capacity of the preference model is equally important to the quality of the LLM-generated preferences. This point has not been previously emphasized or analyzed in past work, and is useful for the broader ML community.
>
>
> >The paper demonstrates the method primarily on environments where progress is relatively obvious (e.g., clear Wordle feedback, discrete game states in NetHack). However, many real-world tasks involve subtle, continuous progress where improvements may be hard to detect from observations. I have several concerns regard this problem:
> -How sensitive is the LLM's preference detection to small state changes?
> -Could the method miss important incremental progress by only detecting large, obvious changes?
>
> We want to emphasize that there are no guarantees that any given LLM will detect important incremental progress for any given environment. It is entirely a function of the LLM's pretraining and the nature of the current task.
>
> From an empirical point of view, the fact that the LLM is distilled into a reward model which can then be used to learn a policy performing well on MetaWorld, Wordle and NetHack suggests that it captures changes essential for value and policy learning, however large or small.
>
>
> We acknowledge that these are important questions and will clearly mention in the paper that the analyzed methods are limited by the ability of an LLM to understand the subtleties of the observation space.

---

> > ### Author Response · Authors · 2024-11-22
> >
> > >Sec 4.1 shows some evidence that AI feedback can help with credit assignment, the underlying mechanism isn't clear.
> >
> > We believe a way to understand the underlying mechanism is through Figure 3. Figure 3 (left) suggests that the LLM-based reward emphasizes key milestones in a trajectory, i.e. implicitly rewarding intermediate sub-goals. This densifies the reward signal, making it easier for an RL agent to learn from its experience as opposed to sparse environment reward (in this experiment, +1 for reaching the green goal). LLM feedback can be thought of as *reward redistribution*, which is a known technique for credit assignment (see the RUDDER method (Arjona-Medina, 2019) for a well-known example).
> >
> > As value functions themselves are based redistributing rewards according to the temporal difference learning rule, our experiments in Figure 4 present a systematic way across domains of evaluating whether LLM feedback redistirbutes rewards (and helps credit assignment).
> >
> > >In complex tasks, progress often results from a sequence of coordinated actions rather than single decisions. Can the method distinguish between critical and auxiliary actions when both contribute to the final outcome?
> >
> > Based on our empirical observations, LLM feedback does not necessarily produce a perfect heuristic that will guide an RL agent every step of the way. LLM feedback seems to be particularly good at highlighting critical milestones (due to the discriminative nature of PBRL), while the RL mechanism (such as temporal difference learning) can attend to the auxiliary actions. That being said, there is no theoretical limitation to the method of LLM feedback that would not allow it f happening: the feedback quality depends on the alignment of the pre-trained LLM to the downstream RL task.
> >
> > ## References
> > 1. Wirth, Christian, et al. "A survey of preference-based reinforcement learning methods." Journal of Machine Learning Research 18.136 (2017): 1-46.
> > 2. Klissarov, Martin, et al. "Motif: Intrinsic motivation from artificial intelligence feedback." arXiv preprint arXiv:2310.00166 (2023).
> > 3. Arjona-Medina, Jose A., et al. "Rudder: Return decomposition for delayed rewards." Advances in Neural Information Processing Systems 32 (2019).

---

> > > ### Author Response · Authors · 2024-11-25
> > >
> > > Dear Reviewer, we appreciate the time you've dedicated to reviewing our submission. With the discussion phase concluding tomorrow, we wanted to respectfully remind you of our responses to your concerns.
> > >
> > > In particular, we have provided answers to the concerns about (1) the potential for inconsistency in the preferences generated by the LLM, (2) provided evidence as to the ability of current LLM to detect small incremental progress, (3) connected the results on credit assignment to the reward redistribution literature, and (4) discussed the ability to distinguish between critical and auxiliary actions.
> > >
> > > We would be grateful if the Reviewer could let us know if they’ve had a chance to read our answers and if we have adequately addressed their points.

---

> > > > ### Comment · Reviewer_9H9A · 2024-11-26
> > > >
> > > > I thank the authors for the response, which has addressed most of my concerns. For now, I will keep my score, and I will also follow the authors' discussion with other reviewers.

---

### Official Review · Reviewer_8WYf · 2024-11-03

**Soundness:** 3
**Presentation:** 3
**Contribution:** 2
**Rating:** 6
**Confidence:** 3

**Summary:**

This paper explores the potential of large language models (LLMs) for tackling complex sequential decision-making problems in reinforcement learning (RL). The authors investigate two approaches: using LLMs to directly generate actions or indirectly generate reward models that guide RL agent training. Their findings demonstrate that even without specialized fine-tuning, LLMs excel at reward modeling. Notably, leveraging AI feedback to craft rewards emerges as a highly effective and generalizable strategy. This approach enhances performance by improving both credit assignment and exploration within the RL framework. Furthermore, the authors address the challenge of unfamiliar environments by demonstrating that fine-tuning LLMs with synthetic data significantly boosts their reward modeling capabilities. Importantly, this fine-tuning process effectively mitigates catastrophic forgetting, preserving the LLM's broad knowledge base.

**Strengths:**

## Pros

- The studied topic is very interesting. The authors explore the sequence modeling capability of large language models in the context of reinforcement learning, which can be of interest to a large number of researchers in the community
- The authors study either directly using LLM as the policy or leveraging LLM for decision-making and conduct numerous experiments
- The authors show that without task-specific fine-tuning, current LLMs only show limited decision-making capabilities when directly generating actions. Furthermore, the authors find that AI-feedback-based rewards produce dense functions that correlate positively with high-quality value functions. Such reward functions can significantly reduce the difficulty of assigning credit by redistributing rewards across different steps within a trajectory. Some of these observations and conclusions are interesting and can be useful for researchers, e.g., using LLM to directly output scalar reward signals can be surprisingly good on some tasks
- The results reported are averaged over 10 seeds, which is great considering the unstable nature of RL

**Weaknesses:**

## Cons

- This paper does not propose any new method and looks more like an empirical paper that investigates the application of LLMs in either direct policy modeling or its capability of facilitating policy learning. Although I agree that such kind of paper is also of importance to the community, its technical novelty is still somewhat weak.
- Some conclusions derived by the authors are not surprising. For example, in environments with unfamiliar dynamics, fine-tuning LLMs with synthetic data can significantly improve their reward modeling capabilities while mitigating catastrophic forgetting. Many of the observations are actually known or acknowledged by many scholars and this paper seems to verify them through some designed experiments
- Another concern is that the conducted experiments may not be necessarily sufficient to back up the observations and conclusions in this paper. I list some of them below
  - the studied topics are limited and some selected topics are too broad to be covered in depth within a 10-page ICLR paper. The considered experimental settings are limited as described in Section 3 (lines 207-215), i.e., the authors only study approaches that can directly interface with the action space supported in the environment and assume that LLMs are used without any gradient update. When dealing with indirect policy learning, the authors only consider reward as code. Yet, there are many other options, e.g., LLMs to execute high-level plans for the agent, etc. Furthermore, the authors only consider a simplified version of reward as code without introducing some extra components or tricks as recent works do [1, 2, 3]. When using LLMs for exploration, the authors do not consider the possibility of letting LLMs write intrinsic reward functions, constructing intrinsic rewards [4, 5], etc.
  - the number of the base LLMs is limited. The authors used the closed-source GPT-4o model for direct policy modeling and the open-source Llama 3 for indirect policy modeling when environment observations consist of text, and PaliGemma when environment observations consist of pixel images. The number of LLMs is quite limited, making it hard to tell the significance and applicability of the observations and conclusions
  - the authors do not compare many baselines in this paper, e.g., the authors only consider the count-based exploration method in Section 4.2 (Figure 5). It would be better to compare against stronger baselines.

[1] Text2reward: Automated dense reward function generation for reinforcement learning

[2] Auto mc-reward: Automated dense reward design with large language models for minecraft

[3]  Eureka: Human-level reward design via coding large language models

[4] Guiding pretraining in reinforcement learning with large language models

[5] World Models with Hints of Large Language Models for Goal Achieving

**Questions:**

- It seems that this work has some preliminary requirements for the LLMs to make it function well, e.g., chain-ot-thought (CoT), in-context learning, and self-refinement, as described in Section 2.1. How important are these components to the sequential decision-making capabilities of the LLMs? How do they affect the direct or indirect policy modeling of LLMs?
- Under many real-world scenarios, we often expect a timely or frequent decision-making ability of the agent. When using LLMs directly or indirectly during the decision-making process, how can we guarantee that the decision is made timely yet effective?
- How much expert data do you use to fine-tune the PaliGemma model in Section 5? How about a different dataset quality, will there be a significant difference? I would expect a more in-depth discussion on fine-tuning LLMs here

---

> ### Author Response · Authors · 2024-11-22
>
> Thank you for your feedback!
>
>
> >This paper does not propose any new method and looks more like an empirical paper that investigates the application of LLMs in either direct policy modeling or its capability of facilitating policy learning. Although I agree that such kind of paper is also of importance to the community, its technical novelty is still somewhat weak.
>
> We thank the reviewer for appreciating our submission's focus on performing unbiased scientific investigations. We totally agree that the community, currently strongly driven by benchmark climbing, could benefit from such work.
>
> We would like to re-iterate some novelties of our work. Section 4 focuses on understanding *why* indirect policy modeling through AI feedback works particularly well. We present qualitative (Figure 3) and quantitative (Figure 4) results that clearly illustrate that one of the main mechanism behind this performance: LLMs naturally improve upon the main challenge of credit assignment in RL. This **has not been previously reported**, except anecdotically in Klissarov et al. 2024, which we built upon as intuition to present concrete scientific evidence. Additionally, in Appendix A.7, we present theoretical connections to the field of heuristic guided reinforcement learning and LLM feedback, and how this relates to credit assignment.
>
> We also investigate how LLM feedback can be used to help in term of exploration, another fundamental challenge in RL. In Figure 5, we show that naively using LLM to promote exploration (i.e. by only changing the prompt) does not work and that careful consideration is needed in terms of (1) continually querying the LLM with the newest experience and (2) leveraging a non-Markovian reward function. **These contributions were not previously reported**, and we believe that they are very important for future work leveraging LLMs for reward modeling.
>
> Finally, Figure 6 presents an important trade-off for practitioners when fine-tuning LLMs for decision making. Direct policy modeling can bring stronger RL agents, but at the cost of catastrophic forgetting. Reward modeling can mitigate this catastrophic, albeit at the cost of potentially lower RL performance. This important result **has also not been previously reported**.
>
>
> >Some conclusions derived by the authors are not surprising. For example, in environments with unfamiliar dynamics, fine-tuning LLMs with synthetic data can significantly improve their reward modeling capabilities while mitigating catastrophic forgetting. Many of the observations are actually known or acknowledged by many scholars and this paper seems to verify them through some designed experiments
>
> We would like to highlight that fine-tuning LLMs with synthetic data *can* have strong adverse effects on their capabilities and **can cause catastrophic forgetting**. In fact, this is one of the main messages of Section 5, where we show that this is the case when LLM fine-tuning is done for policy modeling. What is novel, is that we are showing that **this can be avoided by fine-tuning for reward modeling**, which was not previously reported.
>
>
> >the studied topics are limited and some selected topics are too broad to be covered in depth within a 10-page ICLR paper. The considered experimental settings are limited as described in Section 3 (lines 207-215), i.e., the authors only study approaches that can directly interface with the action space supported in the environment and assume that LLMs are used without any gradient update.
>
> As correctly noted by the reviewer, we introduced a focused and concrete experimental setting for all baselines. Our motivations for studying the capabilities of LLMs to help decision-making without fine-tuning are: (1) training LLMs is computationally expensive and therefore only a handful of labs can afford the process (2) zero-shot or few-shot performance of non-finetuned LLMs on common reasoning benchmarks keeps increasing, meaning that our findings will be increasingly relevant with stronger models. Our results would therefore be of particular interest to a large group of readers.
>
> In Section 5, we then relax the assumption that the LLM is used without fine-tuning and present additional insights when the LLM is fine-tuned. We also include additional results (please see common response) where we compare indirect and direct policy modeling, under the same amount of samples, in terms of performance and compute budget.

---

> > ### Author Response · Authors · 2024-11-22
> >
> > >When dealing with indirect policy learning, the authors only consider reward as code. Yet, there are many other options, e.g., LLMs to execute high-level plans for the agent, etc.
> >
> > We agree with the reviewer and have acknowledged that indirect policy modeling can take many forms (see paragraph between L134 and L140). We also present initial investigations into indirectly modeling the policy through the transition function (Appendix A.4 and Figure 2b). We would like to point out that our LLM Policy baseline (taken from Kim et al, 2024) is first queried to generate a high level plan (just like it is defined in Kim et al, 2024). After obtaining a plan, the policy then proceeds to choose actions step by step.
> >
> > We would be happy to explicitly include a discussion on indirectly modeling the policy through high-level plans and refer to any papers that the reviewers deems appropriate. A high-level plan can be constructed in various ways: ranging from in-context learning to hierarchical reinforcement learning agents. We believe there are many variables to be decided upon. For example, what is the space over which plans are established? Is it the full observation space, a subset of it (e.g., in Minecraft, should it be pixel space or the inventory)? What action space should be considered (e.g., in web agents tasks should it be low level mouse click and drag, or human-crafted high levels as we see in many baselines?)? Additionally, the choice of these variables might not easily generalize across many environments, which is why we decided not to focus on this direction. We do believe that this is a particularly important direction for future work.
> >
> > >Furthermore, the authors only consider a simplified version of reward as code without introducing some extra components or tricks as recent works do [1, 2, 3].
> >
> > Following the reviewer's advice, we have looked into all of the suggested papers to include additional baselines.
> >
> > We believe that one major difference with [1] is the inclusion of human feedback and leveraging RL instead of MPC. As we are already learning policies through RL, the only remaining difference is including human feedback. This choice is quite orthogonal to the idea of using LLM to code rewards, and can arguably be integrated within any approach.
> >
> > It is particularly difficult to generalize [3], as it relies on an extremely fast RL loop implemented in IsaacGym. Without such fast simulators, attempting their approach on another domain is highly impractical as it could take weeks. It also suggests that the approach is not as sample-efficient to the degree that other methods are.
> >
> > [3]’s idea of updating the reward function after interacting with the environment is also seen in [2], which we incorporate as an additional baseline. Specifically, we include a Reward Critic (for syntax/semantic error detection) and a Trajectory Analyzer (to summarize interactions and suggest improvements for the Reward Designer). We report results in this [figure](https://imgur.com/a/PWwUJfz).
> >
> > We notice that updating the reward function with environment interactions provides improvements on MiniWob-Hard, but does not significantly change the performance on the rest of the environments. We emphasize that MiniWob-Hard is the easiest environment out of all the ones we considered (for example the trajectory length of MiniWob-Hard is on average 4 steps, whereas NetHack is about 870 steps).
> >
> > Finally, we would like to highlight that our paper recognizes the strengths of the Reward as Code baseline. In particular, in L261 to L263, we refer the reader to Appendix A.5 where we present under which assumptions can Reward as Code achieve great performance on MetaWorld.
> >
> > In all cases, we will include citations to all of these works in our paper and include the new Reward as Code baseline in the main text.
> >
> > >When using LLMs for exploration, the authors do not consider the possibility of letting LLMs write intrinsic reward functions, constructing intrinsic rewards [4, 5], etc.
> >
> > To the best of our knowledge, [4, 5] leverage the similarity between a language goal description and the current observation (which, for example, can be a caption of an RGB frame). In fact, we refer to [4] in our paper as the method on which we build our "Embedding-based" baseline (we call it embedding based as the cosine similarity of measured between embeddings). We are happy to mention [5] in our paper as well as the paper seems to closely follow the setup of [4] in terms of defining the reward. We are also happy to refer to these works in Section 4.2, although the embedding-based baseline does not perform very well on the hard exploration domain of NetHack.

---

> > > ### Author Response · Authors · 2024-11-22
> > >
> > > >the number of the base LLMs is limited. The authors used the closed-source GPT-4o model for direct policy modeling and the open-source Llama 3 for indirect policy modeling when environment observations consist of text, and PaliGemma when environment observations consist of pixel images. The number of LLMs is quite limited, making it hard to tell the significance and applicability of the observations and conclusions
> > >
> > > We perform additional experiments for the direct policy modeling method, using Gemini-Experimental-1114 and Claude 3.5 Sonnet. We also additionally investigate the performance of Qwen 2.5 Instruct for the AI Feedback baseline on all text-based environemnts (where we perviously had Llama 3), and investigate the performance of Llama 3.2-V 11b on MetaWorld (where we previously had PaliGemma). We report results for [direct policy modeling](https://imgur.com/a/vlCNtMV) and [indirect policy modeling](https://imgur.com/a/DMu5dZ1). We notice that the performance for direct policy modeling approaches does not fluctuate significantly with respect to the model being used. The performance of indirect policy modeling with AI Feedback is also generally similar, with the exception of Llama 3.2-V being significantly better than PaliGemma.
> > >
> > > These results are perhaps unsurprising as most LLMs follow very similar training principles, as can be seen through the results posted across many public benchmarks. The biggest difference in performance comes between the two multi-modal LLMs (Llama 3.2-V and PaliGemma), which is an active area research. We now in total have 7 LLMs over 4 different domains, covering a variety of policy modeling approaches.
> > >
> > > >the authors do not compare many baselines in this paper, e.g., the authors only consider the count-based exploration method in Section 4.2 (Figure 5). It would be better to compare against stronger baselines.
> > >
> > > We would like to highlight that we evaluate 5 methods in total (including direct and indirect policy modeling) in the main paper and we do so across a wide variety of environments. Most RL papers do not conduct such wide investigation, and most of the times LLM+RL paper present only few seeds (we present 10, which is, thankfully, noted by the reviewer). We have also added 7 additional baselines in the rebuttal itself.
> > >
> > > For the precise comment on additional baselines in Figure 5, we would like to clarify that the counting function is used within the LLM-based rewards shown in Equation 4. A count-based baseline is not presented in this work. We would however like to invite the reviewer to look at the previous work of [5] at their Figure 7 from Appendix F. In these figures, the authors show that their approach, which is based on AI feedback (as is ours), significantly outperforms previous exploration baselines. We will include this discussion in our submission.
> > >
> > > >It seems that this work has some preliminary requirements for the LLMs to make it function well, e.g., chain-ot-thought (CoT), in-context learning, and self-refinement, as described in Section 2.1. How important are these components to the sequential decision-making capabilities of the LLMs? How do they affect the direct or indirect policy modeling of LLMs?
> > >
> > > We agree that this is an important question and we would like to refer the reviewer to Appendix A.3 and Figure 7 where we provide answers for the direct policy modeling method. In essence, all these techniques are particularly important to get the best performance for direct policy modeling.
> > >
> > > When it comes to indirect policy modeling, all the experiments are reported without in-context learning and without self-refinement. To study the impact of CoT on indirect policy modeling, we have changed the prompt to the following:
> > >
> > > ```
> > > I will present you with two short gameplay descriptions of Nethack.
> > > Express a preference based on which description is the most likely to make some progress towards the goal, writing either ("best_description":  1), ("best_description":  2).
> > > You could also say ("best_description":  None).
> > > description_1:  {description_1}
> > > description_2:  {description_2}
> > > ```
> > > which is the same as the original prompt used (see Appendix A.2), except that we have removed any request to refelect on the descriptions. We report results in this [figure](https://imgur.com/a/2m6odah), which shows no statistical difference in performance compared to using CoT. To understand this result, we can refer to [6] which shows that across 14 LLMs, CoT only significantly helps for mathematical and symbolic problems. We believe the way LLMs help the tasks studied in our submission falls under the Commonsense and Knowledge categories of [6]. We will include these results in our paper.

---

> > > > ### Author Response · Authors · 2024-11-22
> > > >
> > > > >Under many real-world scenarios, we often expect a timely or frequent decision-making ability of the agent. When using LLMs directly or indirectly during the decision-making process, how can we guarantee that the decision is made timely yet effective?
> > > >
> > > > This is a very important question, which we believe highlights another trade-off between direct and indirect policy modeling.
> > > >
> > > > Direct policy modeling implies that the LLM is executed in the environment, which might limit the frequency with which we can query it (for example, the $\pi_0$ report from Black et al. 2024 emphasizes the necessity of using special techniques such as action chunking to achieve a 50 Hz control frequency). In contrast, indirect policy modeling can distill an LLM's knowledge into a smaller neural network.  Such network would be queried much faster, and the resulting robot could react more promptly to situations.
> > > >
> > > > However, direct policy modeling can leverage in-context learning to adapt faster to new situations, whereas indirect policy modeling relies on gradient updates for learning. As LLMs become better at in-context learning , it is possible that it would become reliable enough to be deployed in real world scenarios.
> > > >
> > > > We are happy to include this discussion in the revised version of the paper.
> > > >
> > > >
> > > > >How much expert data do you use to fine-tune the PaliGemma model in Section 5? How about a different dataset quality, will there be a significant difference? I would expect a more in-depth discussion on fine-tuning LLMs here
> > > >
> > > > **Indirect policy modeling fine-tuning**: Only 100 datapoints were used. The data itself is not generated by an expert: it is the data that was generated by the RL agent obtaining a success rate of ~0.15 on the task. This closely mimicks obtaining a policy improvement step from suboptimal data as seen in classical RL formulations.
> > > >
> > > > **Direct policy modeling fine-tuning**: It relies on 90k observation-action transitions from a scripted expert with 100% success rate. It is likely that a dataset with low quality trajectories will have a strong impact as the fine-tuning is done through imitation learning. See Kumar et al (2022) for an in-depth analysis of when to learn from optimal vs suboptimal data.
> > > >
> > > > We believe these results show the generality of the indirect policy modeling approach, where fine-tuning can be done with less restrictive assumptions on the data.
> > > >
> > > > ## References
> > > > 1. Kim, Geunwoo, Pierre Baldi, and Stephen McAleer. "Language models can solve computer tasks." Advances in Neural Information Processing Systems 36 (2024).
> > > > 2. Yu, Wenhao, et al. "Language to rewards for robotic skill synthesis." arXiv preprint arXiv:2306.08647 (2023).
> > > > 3. https://www.physicalintelligence.company/download/pi0.pdf
> > > > 4. Kumar, Aviral, et al. "When should we prefer offline reinforcement learning over behavioral cloning?." arXiv preprint arXiv:2204.05618 (2022).
> > > > 5. Klissarov, Martin, et al. "Motif: Intrinsic motivation from artificial intelligence feedback." arXiv preprint arXiv:2310.00166 (2023).
> > > > 6. Sprague, et al. "To CoT or not to CoT? Chain-of-thought helps mainly on math and symbolic reasoning"(2024).

---

> > > > > ### Author Response · Authors · 2024-11-25
> > > > >
> > > > > Dear Reviewer, we appreciate the time you've dedicated to reviewing our submission. With the discussion phase concluding tomorrow, we wanted to respectfully remind you of our responses to your concerns.
> > > > >
> > > > > In particular, we have provided answers to the concerns about (1) novelty by highlighting some of the contributions that are reported for the first time in our submission, (2) the experimental setting by running additional experiments/baselines around LLM fine-tuning which keep track of samples and FLOPS, (3) added a discussion about leveraging LLMs for generating plans, (4) conducted experiments with the requested baseline for Reward as Code and added multiple new baselines by leveraging various LLMs, (5) performed a study on the impact of CoT, (6) added a discussion around the trade-offs in real world scenarios, and (7) provided more details about how fine-tuning was performed.
> > > > >
> > > > > We would be grateful if the Reviewer could let us know if they’ve had a chance to read our answers and if we have adequately addressed their points.

---

> > > > > > ### Comment · Reviewer_8WYf · 2024-11-26
> > > > > >
> > > > > > Thanks for the detailed rebuttal. Despite that I still think that the technical novelty of this paper is limited, I agree that this paper presents some interesting observations. I appreciate that the authors conduct numerous experiments to address my concerns. I have adjusted my score accordingly.
> > > > > >
> > > > > > A minor suggestion: the authors can consider directly including the new experiments in the revision since this venue allows a revision during the rebuttal period.

---

> > > > > > > ### Author Response · Authors · 2024-11-28
> > > > > > >
> > > > > > > We thank the Reviewer for their answer and for being open to the contributions of our paper. We have just updated the PDF with all the additional experiments and discussions. We are currently running the LLM fine-tuning experiment, measuring FLOPS, samples and success rate, on NetHack, and will soon after run it on MetaWorld.

---

### Official Review · Reviewer_3Jq6 · 2024-11-04

**Soundness:** 1
**Presentation:** 3
**Contribution:** 1
**Rating:** 5
**Confidence:** 4

**Summary:**

This paper studied the capabilities of Large Language Models (LLMs) for reinforcement learning (RL) in interactive decision-making tasks, by directly producing policies or indirectly generating rewards for policy optimization.

**Strengths:**

1. The paper is well-written and easy to follow.
2. The experiments are conducted on various environments.

**Weaknesses:**

1. The motivation of this paper is limited. One conclusion of the paper is that leveraging the feedback from the LLM to optimize the policy is better than using the LLM directly as a policy. However, since the latter is fine-tuning free, it is not surprising that the former is better. Even if the AI feedback is noisy, as long as the overall feedback is accurate, fine-tuning the policy against it can obtain some improvements.
2. The policy modeling and reward modeling abilities of LLMs in decision-making tasks have been widely studied in previous works [1,2]. No new methods are proposed in this paper, and the novelty is thus limited compared to these works.
3. The analysis is also not convincing. The authors claim that AI feedback helps better credit assignments. However, given only the training curves, it is not clear when and why credit is better assigned.

[1] Yao et al., ReAct: Synergizing Reasoning and Acting in Language Models.\
[2] Ma et al., Eureka: Human-Level Reward Design via Coding Large Language Models.

**Questions:**

See weakness.

---

> ### Author Response · Authors · 2024-11-22
>
> >The motivation of this paper is limited. One conclusion of the paper is that leveraging the feedback from the LLM to optimize the policy is better than using the LLM directly as a policy. However, since the latter is fine-tuning free, it is not surprising that the former is better.
>
>
> One of the goals of the paper is to understand how LLMs can represent the policy, reward or transition function. Our motivations for studying these capabilities without fine-tuning are: (1) training LLMs is computationally expensive and therefore only a handful of labs can afford the process (2) zero-shot or few-shot performance of non-finetuned LLMs on common reasoning benchmarks keeps increasing, meaning that our findings will be increasingly relevant with stronger models. Our results would therefore be of particular interest to a large group of readers. In Section 5, we then relax the assumption that the LLM is used without fine-tuning and present additional insights when the LLM is fine-tuned.
>
> To address the reviewer's point, we include the following results to give a complete picture of the LLM capabilities for decision-making. In the tables below, we directly compare the  performance of direct policy modeling and indirect policy modeling (AI Feedback) on one of the MetaWorld tasks. In these results, we control both for the amount of samples and the amount of FLOPS. We compute the FLOPs using the package [calflops](https://github.com/MrYxJ/calculate-flops.pytorch), whose results we found consistent with Jiang et al (2024).
>
> Specifically, we fine-tune the direct policy method under three settings: only with online RL, only with SFT, and online RL training on an SFT-pretrained model. We compare these baselines to the indirect policy modeling approach of AI Feedback that uses RL for learning.
>
>
> Results indicate that indirect policy modeling achieves comparable performance to direct policy modeling trained with SFT at a fraction of the computational cost, i.e. two orders of magnitude lower. Policy modeling achieves a higher success rate with 1M training samples, but is otherwise significantly less efficient than indirect policy modeling. Importantly, direct policy modeling with SFT relies on a large expert dataset to achieve good performance. On the other hand, direct policy modeling trained only with RL shows poor performance with 1M samples and high sensitivity to hyperparameters. This is in line with recent work (under anonymous submission) reporting  low sample efficiency when training LLMs with online RL, *without* SFT, for sequential decision-making.
>
>
> ### Success rate
> | Environment Samples | Direct Policy (only RL) | Direct Policy (only SFT) | Direct Policy (RL after SFT pretaining) | Indirect Policy |
> | -------- | -------- | -------- | -------- | -------- |
> | 0    | 0        | 0        | 0.92     | 0         |
> | 128k | 0.01     | 0.10     | 0.80     |  0.22    |
> | 256k | 0.05     | 0.13     | 0.63     | 0.48     |
> | 1M | 0.09     | 0.87     | 0.65     | 0.72     |
>
> ### FLOPS
> | Environment Samples | Direct Policy (only RL) | Direct Policy (RL after SFT pretaining) | Indirect Policy |
> | -------- | -------- | -------- | -------- |
> | 0 |  0    |  3.65x10^17    | 6.9x10^12     |
> | 128k |  7.36x10^16    | 4.38x10^17     | 4.09×10^14     |
> | 256k | 1.46x10^17     | 5.11x10^17     | 8.109×10^14     |
> | 1M | 7.12x10^17     | 1.09x10^18     | 3.145x10^15     |
>
>
> We will add these results in a revised version of paper and extend the comparison to the rest of the domains given enough time to run the experiments.

---

> > ### Author Response · Authors · 2024-11-22
> >
> > >The policy modeling and reward modeling abilities of LLMs in decision-making tasks have been widely studied in previous works [1,2]. No new methods are proposed in this paper, and the novelty is thus limited compared to these works.
> >
> > It is hard to draw overarching conclusions from [1,2] as they do not provide an apples-to-apples comparison, each claim state-of-the-art under very different experimental settings. We propose to address this mismatch by evaluating, across a wide range of environments and under similar conditions, the capabilities of LLMs for decision-making when the LLMs are not fine-tuned, and when they are fine-tuned (Section 3 and 5), and discuss trade-offs across the different choices. We present hypotheses as to why some methods are working better than others, based on fundamental concepts in RL (Section 3 and Section 4).
> >
> > >The analysis is also not convincing. The authors claim that AI feedback helps better credit assignments. However, given only the training curves, it is not clear when and why credit is better assigned.
> >
> > We would like to point the reviewer's attention to Section 4, where we present both qualitative and quantitative results to support this hypothesis. Figure 3 shows a clear example of when and how credit assignment is done in an easily understandable environment. In particular, the LLM reward emphasizes important moments such as picking up the key and opening the door. This is aso known as *reward redistribution*, which is a common way of performing creedit assignment (see Arjona-Medina (2019) as an example)
> >
> > Figure 4 then presents the correlation between the reward model obtained from the LLM and the value function obtained from training an RL agent on environment reward. This figure shows the LLM-based reward model strongly correlates with value functions obtained later in training. This strong correlation indicates that LLMs naturally distribute credit throughout trajectories.
> >
> > ## References
> > 1. Arjona-Medina, Jose A., et al. "Rudder: Return decomposition for delayed rewards." Advances in Neural Information Processing Systems 32 (2019).
> > 2. Schulman, John, et al. "High-dimensional continuous control using generalized advantage estimation." arXiv preprint arXiv:1506.02438 (2015).

---

> > > ### Author Response · Authors · 2024-11-25
> > >
> > > Dear Reviewer, we appreciate the time you've dedicated to reviewing our submission. With the discussion phase concluding tomorrow, we wanted to respectfully remind you of our responses to your concerns.
> > >
> > > In particular, we have provided answers to the concerns about (1) the experimental setting by running additional experiments/baselines around LLM fine-tuning which keep track of samples and FLOPS, (2) provided context to the reason of this study, which is significantly different than any previous work, and (3) explained in more detail the credit assignment mechanism, citing a well-known line of research around reward redistribution.
> > >
> > > We would be grateful if the Reviewer could let us know if they’ve had a chance to read our answers and if we have adequately addressed their points.

---

> > > > ### Comment · Reviewer_3Jq6 · 2024-11-26
> > > > **Thanks for the rebuttal**
> > > >
> > > > I thank the authors for the responses. My concern regarding the fair comparison between policy modeling and reward modeling is addressed. For this reason, I raised my score to 5. However, I still think the technical novelty is somewhat limited, with no new method proposed, and fails to provide enough in-depth experiment analysis. For example, my concern regarding why modeling rewards help better credit assignments than modeling policy still remains. Since the authors claim they provide apples-to-apples comparisons, an example of important moments during reward learning is not enough to demonstrate why policy modeling is worse. Some statistical analysis and large-scale experiments are necessary to draw this conclusion.

---

> > > > > ### Author Response · Authors · 2024-11-28
> > > > >
> > > > > We thank the Reviewer for taking the time to continue the discussion.
> > > > >
> > > > > We believe there might be a misunderstanding: we make no claim that reward modelling performs better credit assignment than policy modelling. The results of Section 4 are meant to understand how the LLM-based reward modelling leads to good RL results. We have added an additional sentence at the beginning of Section 4 to make this as clear as we can. If there are moments in Section 4 that create this misunderstanding, we kindly ask the Reviewer to share them so we can address them.
> > > > >
> > > > > Concerning novelty we believe that novelty of a paper can be measured by other ways than suggesting a new method. In our case, this can be observed by a large set of diverse experiments, across multiple domains and LLMs, studying important unanswered questions pertaining to using LLMs in RL. We list some of these contributions as concisely as we can here:
> > > > > - Investigating direct vs indirect policy modelling without LLM fine-tuning (Figure 1 and 2a) and now with LLM fine-tuning (experiments from rebuttal).
> > > > > - Studying how LLMs naturally improve credit assignment in RL, supported by qualitative (Figure 3) and quantitative (Figure 4) results.
> > > > > - Establishing theoretical links between LLM feedback, heuristic-guided reinforcement learning, and credit assignment (Appendix A.7).
> > > > > - Showing that naive use of LLMs for exploration fails and highlights the importance of continual querying and non-Markovian reward functions (Figure 5).
> > > > > - Identifying a trade-off between fine-tuning LLMs for direct policy modelling (higher RL performance but catastrophic forgetting) and reward modeling (mitigates forgetting but reduces RL performance) (Figure 6).
> > > > >
> > > > > These results are supplemented by further experiments in the Appendix which provide further context and points to potential future work. We believe these insights significantly advance the understanding of LLMs in RL, and we hope they resonate with the Reviewer.

---

> > > > > > ### Author Response · Authors · 2024-12-02
> > > > > >
> > > > > > Dear Reviewer, we hope our previous message has clarified the misunderstanding with regards to the credit assignment experiments.
> > > > > >
> > > > > > As promised, we follow-up with experimental results on Nethack, which show that indirect policy modeling is $\times 10$ more FLOPS-efficient and $\times 5$ more sample-efficient than direct policy modeling during online RL training. Overall, the results continue to support the conclusions of our submission.
> > > > > >
> > > > > > | Environment Samples | Direct Policy Modeling (FLOPS) | Direct Policy Modeling (Success Rate) |  Indirect Policy Modeling (FLOPS) | Indirect Policy Modeling (Success Rate) |
> > > > > > | -------- | -------- | -------- |---|---|
> > > > > > | 0 |  0    | 0.0% | 4.62 x 10^18| 0.0%|
> > > > > > | 10M |  7.12x10^18    | 3.1% |4.69 x 10^18| 12.4%|
> > > > > > | 50M |   3.56x10^19   | 5.3% |4.97 x 10^18|31.1%|
> > > > > > | 100M | 7.12x10^19     | 6.9% |5.32 x 10^18|39.7%|

---

### Official Review · Reviewer_57NJ · 2024-11-06

**Soundness:** 3
**Presentation:** 3
**Contribution:** 2
**Rating:** 5
**Confidence:** 3

**Summary:**

In this paper, the authors study how Large Language Models (LLMs) can produce decision-making policies, either by generating actions or by creating reward models for reinforcement learning (RL). The authors use experimental results to reveal that LLMs excel at reward modeling, particularly when using AI feedback, and fine-tuning with synthetic data enhances performance in unfamiliar environments, helping prevent catastrophic forgetting.

**Strengths:**

1. The authors perform extensive experiments to evaluate the capabilities of large language models (LLMs) in sequential decision-making tasks. Specifically, they demonstrate that using LLMs to generate preference data and then train a reward model for RL training has the highest performance gain compared with other methods such as directly modeling policy by LLMs. This conclusion seems interesting and helpful for future works in designing RL agents incorporating LLMs.

2. The experiments in Fig. 3 and Fig. 4 clearly illustrate the effect of the rewards learned through the LLM Feedback.

3. It is interesting that the authors show that prompting engineering can steer LLMs for active exploration on the
NetHack task, which approximates the count-based exploration bonus.

**Weaknesses:**

1. The related works section is very incomplete. The authors should discuss more recent works that study LLM decision-making problems and self-rewarding of LLMs such as [1]-[7]. The introduction of LLM for better reward design on RL is also studied in [8] and should be discussed carefully.

2. The authors compare direct and indirect policy modeling and use experiments to show that indirect policy modeling attains higher performance in multiple tasks. However, I am a little unconvinced about the experimental configuration, where the authors only query the GPT for the direct policy modeling but train an RL agent with multiple steps for the indirect policy modeling. It seems that the computation cost and the number of interactions with the environment are not controlled. Moreover, it has been shown in ([1]-[3]) that LLM can also serve as a critic model to give a numerical estimated value in the direct policy modeling. Maybe the authors should also add an experiment to analyze what would happen if the estimated reward from the indirect policy modeling were incorporated into the direct policy modeling. In this case, no RL training is involved and computation cost is easier to control.

[1] Zhou, Andy, et al. "Language agent tree search unifies reasoning acting and planning in language models." arXiv preprint arXiv:2310.04406 (2023).

[2] Sun, Haotian, et al. "Adaplanner: Adaptive planning from feedback with language models." Advances in Neural Information Processing Systems 36 (2024).

[3] Liu, Zhihan, et al. "Reason for future, act for now: A principled architecture for autonomous llm agents." Forty-first International Conference on Machine Learning. 2023.

[4] Huang, Jen-tse, et al. "How Far Are We on the Decision-Making of LLMs? Evaluating LLMs' Gaming Ability in Multi-Agent Environments." arXiv preprint arXiv:2403.11807 (2024).

[5] Park, Chanwoo, et al. "Do llm agents have regret? a case study in online learning and games." arXiv preprint arXiv:2403.16843 (2024).

[6] Nottingham, Kolby, et al. "Do embodied agents dream of pixelated sheep: Embodied decision making using language guided world modelling." International Conference on Machine Learning. PMLR, 2023.

[7] Yuan, Weizhe, et al. "Self-rewarding language models." arXiv preprint arXiv:2401.10020 (2024).

[8] Kwon, Minae, et al. "Reward design with language models." arXiv preprint arXiv:2303.00001 (2023).

3. Minor typos: Line 762: "except fro".

**Questions:**

1. When reporting the final results in Fig 1, Fig 2, and Fig 6, do the authors control the number of interactions with the environment? I'm also interested in understanding how performance changes with an increasing number of interactions, as shown in the learning curves typically found in reinforcement learning (RL) papers.

---

> ### Author Response · Authors · 2024-11-22
>
> Thank you for your feedback!
>
>
> >The related works section is very incomplete. The authors should discuss more recent works that study LLM decision-making problems and self-rewarding of LLMs such as [1]-[7].
>
> We thank the reviewer for suggesting a set of related works, we will discuss them in-depth in our paper. We briefly highlight some of the connections to our submission here.
>
> Ideas from [1] and [3], such as integrating the critic of an LLM within the decision making process of another LLM, are discussed later in our rebuttal. The resulting performance of [2] on MiniWob-Hard tasks is around $61 \pm 3$ using GPT-4o, which is comparable to RCI (our LLM Policy baseline). [4] focuses on the multi-agent setting and [5] on the notion of regret - both different settings from ours, which we will nevertheless mention in the related works. In [6], the authors use an LLM to provide a plan over MineCraft goals, which is then used within an RL loop. The idea of crafting a plan is within the RCI method (our LLM Policy baseline), albeit executed differently. The work of [7] focuses on the domain of Math and uses an LLM-as-a-Judge with DPO to improve upon the LLM's previous performance. Although quite promising, there is a lot of work to be done to evaluate [7] in a sequential decision-making setting, in particular under long horizons.
>
>
> >The introduction of LLM for better reward design on RL is also studied in [8] and should be discussed carefully.
>
> We are unfortunately unable to find what [8] is referring to.

---

> > ### Comment · Reviewer_57NJ · 2024-11-26
> > **Response to the Authors**
> >
> > Thank the authors for their responses. I have updated the review to show the reference [8].
> >
> > [8] Kwon, Minae, et al. "Reward design with language models." arXiv preprint arXiv:2303.00001 (2023)

---

> > > ### Author Response · Authors · 2024-11-28
> > >
> > > We thank the Reviewer for updating the reference list. We have just updated the PDF with all the additional experiments, related works and key discussions. We are currently running the LLM fine-tuning experiment, measuring FLOPS, samples and success rate, on NetHack, and will soon after run it on MetaWorld.
> > >
> > > We would greatly appreciate it if the Reviewer could share their reactions to the rebuttal and let us know if our additions have adequately addressed their concerns.

---

> > > > ### Comment · Reviewer_57NJ · 2024-12-02
> > > > **Response to the Authors**
> > > >
> > > > Thank the authors for addressing some of my concerns. I appreciate the works done by the authors and increase the 'sound' score in my review from 2 (fair) to 3 (good). However, I am not still fully convinced of this paper's novelty and contribution compared with the related works mentioned in my previous review, hence I would keep my original score for the recommendation.

---

> > > > > ### Author Response · Authors · 2024-12-02
> > > > >
> > > > > As promised, we follow-up with experimental results on Nethack, which show that indirect policy modeling is $\times 10$ more FLOPS-efficient and $\times 5$ more sample-efficient than direct policy modeling during online RL training.
> > > > >
> > > > > | Environment Samples | Direct Policy Modeling (FLOPS) | Direct Policy Modeling (Success Rate) |  Indirect Policy Modeling (FLOPS) | Indirect Policy Modeling (Success Rate) |
> > > > > | -------- | -------- | -------- |---|---|
> > > > > | 0 |  0    | 0.0% | 4.62 x 10^18| 0.0%|
> > > > > | 10M |  7.12x10^18    | 3.1% |4.69 x 10^18| 12.4%|
> > > > > | 50M |   3.56x10^19   | 5.3% |4.97 x 10^18|31.1%|
> > > > > | 100M | 7.12x10^19     | 6.9% |5.32 x 10^18|39.7%|
> > > > >
> > > > >
> > > > > We believe that our work addresses important unanswered research questions through a novel perspective, both in terms of breadth and depth. We list some of these contributions below:
> > > > > - Investigating direct vs indirect policy modelling without LLM fine-tuning (Figure 1 and 2a) and now with LLM fine-tuning (experiments from rebuttal).
> > > > > - Studying how LLMs naturally improve credit assignment in RL, supported by qualitative (Figure 3) and quantitative (Figure 4) results.
> > > > > - Establishing theoretical links between LLM feedback, heuristic-guided reinforcement learning, and credit assignment (Appendix A.7).
> > > > > - Showing that naive use of LLMs for exploration fails and highlighting the importance of continually querying the LLM together with using non-Markovian reward functions (Figure 5).
> > > > > - Identifying a trade-off between fine-tuning LLMs for direct policy modelling (highest RL performance at the cost of catastrophic forgetting) and reward modeling (mitigates forgetting and keeps good RL performance) (Figure 6).
> > > > >
> > > > > Overall, our work provides prescriptive results that significantly advance the understanding of LLMs in RL and has the potential to be particularly useful to the broader research community.
> > > > >
> > > > > At the best of our knowledge, all of these contributions are novel.

---

> ### Author Response · Authors · 2024-11-22
>
> >I am a little unconvinced about the experimental configuration, where the authors only query the GPT for the direct policy modeling but train an RL agent with multiple steps for the indirect policy modeling. It seems that the computation cost and the number of interactions with the environment are not controlled.
>
>
> One of the goals of the paper is to understand how LLMs can represent the policy, reward or transition function. Our motivations for studying these capabilities without fine-tuning are: (1) training LLMs is computationally expensive and therefore only a handful of labs can afford the process (2) zero-shot or few-shot performance of non-finetuned LLMs on common reasoning benchmarks keeps increasing, meaning that our findings will be increasingly relevant with stronger models. Our results would therefore be of particular interest to a large group of readers. In Section 5, we then relax the assumption that the LLM is used without fine-tuning and present additional insights when the LLM is fine-tuned.
>
> To address the reviewer's point, we include the following results to give a complete picture of the LLM capabilities for decision-making. In the tables below, we directly compare the  performance of direct policy modeling and indirect policy modeling (AI Feedback) on one of the MetaWorld tasks. In these results, we control both for the amount of samples and the amount of FLOPS. We compute the FLOPs using the package [calflops](https://github.com/MrYxJ/calculate-flops.pytorch), whose results we found consistent with Jiang et al (2024).
>
> Specifically, we fine-tune the direct policy method under three settings: only with online RL, only with SFT, and online RL training on an SFT-pretrained model. We compare these baselines to the indirect policy modeling approach of AI Feedback that uses RL for learning.
>
>
> Results indicate that indirect policy modeling achieves comparable performance to direct policy modeling trained with SFT at a fraction of the computational cost, i.e. two orders of magnitude lower. Policy modeling achieves a higher success rate with 1M training samples, but is otherwise significantly less efficient than indirect policy modeling. Importantly, direct policy modeling with SFT relies on a large expert dataset to achieve good performance. On the other hand, direct policy modeling trained only with RL shows poor performance with 1M samples and high sensitivity to hyperparameters. This is in line with recent work (under anonymous submission) reporting  low sample efficiency when training LLMs with online RL, *without* SFT, for sequential decision-making.
>
>
> ### Success rate
> | Environment Samples | Direct Policy (only RL) | Direct Policy (only SFT) | Direct Policy (RL after SFT pretaining) | Indirect Policy |
> | -------- | -------- | -------- | -------- | -------- |
> | 0    | 0        | 0        | 0.92     | 0         |
> | 128k | 0.01     | 0.10     | 0.80     |  0.22    |
> | 256k | 0.05     | 0.13     | 0.63     | 0.48     |
> | 1M | 0.09     | 0.87     | 0.65     | 0.72     |
>
> ### FLOPS
> | Environment Samples | Direct Policy (only RL) | Direct Policy (RL after SFT pretaining) | Indirect Policy |
> | -------- | -------- | -------- | -------- |
> | 0 |  0    |  3.65x10^17    | 6.9x10^12     |
> | 128k |  7.36x10^16    | 4.38x10^17     | 4.09×10^14     |
> | 256k | 1.46x10^17     | 5.11x10^17     | 8.109×10^14     |
> | 1M | 7.12x10^17     | 1.09x10^18     | 3.145x10^15     |
>
>
> We will add these results in a revised version of paper and extend the comparison to the rest of the domains given enough time to run the experiments.

---

> > ### Author Response · Authors · 2024-11-22
> >
> > >Moreover, it has been shown in ([1]-[3]) that LLM can also serve as a critic model to give a numerical estimated value in the direct policy modeling. Maybe the authors should also add an experiment to analyze what would happen if the estimated reward from the indirect policy modeling were incorporated into the direct policy modeling. In this case, no RL training is involved and computation cost is easier to control.
> >
> > We have diligently looked into all three suggested papers. For paper [2], it seems like the algorithm is not using the LLM to serve as a critic model to give a numerical estimated value.
> >
> >
> > In [1] and [3], the proposed algorithms use LLM evaluation but rely on MCTS for action selection, unrolling decision trees via an environment simulator (e.g., see this [line](https://github.com/lapisrocks/LanguageAgentTreeSearch/blob/main/webshop/lats.py#L612) in the code of [1]). This dependence on simulators for rollouts limits general applicability (Co-Reyes, 2020). Without a simulator or one that supports resets (common cases), a model must be learned, posing challenges, especially in partially observable environments (Icarte et al., 2023; Hafner et al., 2023). In contrast, our methods are broadly applicable, functioning with simulators, world models, or real-world systems.
> >
> >
> > We conducted an experiment where LLM-based rewards were given to an LLM Policy during online interaction with the environment. After observing a new state, the reward was provided as context for in-context learning, guiding the generation of the next action. Using GPT4o, we observed no significant performance improvement (see [figure](https://imgur.com/a/822YgtT)), with a slight decrease in NetHack. This suggests LLMs struggle to predict the causal effects of actions, limiting their utility--this is essentially our hypothesis at the end of Section 3. Adding LLM-based rewards, though valuable in indirect policy modeling, did not mitigate this limitation.
> >
> >
> > >When reporting the final results in Fig 1, Fig 2, and Fig 6, do the authors control the number of interactions with the environment? I'm also interested in understanding how performance changes with an increasing number of interactions, as shown in the learning curves typically found in reinforcement learning (RL) papers.
> >
> > Please see the above additional results.
> >
> > ## References
> > 1. Co-Reyes, John D., et al. "Ecological reinforcement learning." arXiv preprint arXiv:2006.12478 (2020).
> > 2. Icarte, Rodrigo Toro, et al. "Learning reward machines: A study in partially observable reinforcement learning." Artificial Intelligence 323 (2023): 103989.
> > 3. Hafner, Danijar, et al. "Mastering diverse domains through world models." arXiv preprint arXiv:2301.04104 (2023).
> > 4. Jiang, Chaoya, et al. "MaVEn: An Effective Multi-granularity Hybrid Visual Encoding Framework for Multimodal Large Language Model." arXiv preprint arXiv:2408.12321 (2024).

---

> ### Author Response · Authors · 2024-11-25
>
> Dear Reviewer, we appreciate the time you've dedicated to reviewing our submission. With the discussion phase concluding tomorrow, we wanted to respectfully remind you of our responses to your concerns.
>
> In particular, we have provided answers to the concerns about (1) related work, by including broader discussions and an additional baseline, (2) the experimental setting by running additional experiments/baselines around LLM fine-tuning which keep track of samples and FLOPS, and (3) provided the requested additional baseline.
>
> We would be grateful if the Reviewer could let us know if they’ve had a chance to read our answers and if we have adequately addressed their points.

---

### Author Response · Authors · 2024-11-22

We would like to thank the reviewers for their time spent attending our submission.

We would like to highlight the following two points, which are mentioned in the individual rebuttals.

**LLM Fine-tuning**

One of the goals of the paper is to understand how LLMs can represent the policy, reward or transition function. Our motivations for studying these capabilities without fine-tuning are: (1) training LLMs is computationally expensive and therefore only a handful of labs can afford the process (2) zero-shot or few-shot performance of non-finetuned LLMs on common reasoning benchmarks keeps increasing, meaning that our findings will be increasingly relevant with stronger models. Our results would therefore be of particular interest to a large group of readers. In Section 5, we then relax the assumption that the LLM is used without fine-tuning and present additional insights when the LLM is fine-tuned.

We include the following results to give a complete picture of the LLM capabilities for decision-making. In the tables below, we directly compare the  performance of direct policy modeling and indirect policy modeling (AI Feedback) on one of the MetaWorld tasks. In these results, we control both for the amount of samples and the amount of FLOPS. We compute the FLOPs using the package [calflops](https://github.com/MrYxJ/calculate-flops.pytorch), whose results we found consistent with Jiang et al (2024).

Specifically, we fine-tune the direct policy method under three settings: only with online RL, only with SFT, and online RL training on an SFT-pretrained model. We compare these baselines to the indirect policy modeling approach of AI Feedback that uses RL for learning.


Results indicate that indirect policy modeling achieves comparable performance to direct policy modeling trained with SFT at a fraction of the computational cost, i.e. two orders of magnitude lower. Policy modeling achieves a higher success rate with 1M training samples, but is otherwise significantly less efficient than indirect policy modeling. Importantly, direct policy modeling with SFT relies on a large expert dataset to achieve good performance. On the other hand, direct policy modeling trained only with RL shows poor performance with 1M samples and high sensitivity to hyperparameters. This is in line with recent work (under anonymous submission) reporting  low sample efficiency when training LLMs with online RL, *without* SFT, for sequential decision-making.


### Success rate
| Environment Samples | Direct Policy (only RL) | Direct Policy (only SFT) | Direct Policy (RL after SFT pretaining) | Indirect Policy |
| -------- | -------- | -------- | -------- | -------- |
| 0    | 0        | 0        | 0.92     | 0         |
| 128k | 0.01     | 0.10     | 0.80     |  0.22    |
| 256k | 0.05     | 0.13     | 0.63     | 0.48     |
| 1M | 0.09     | 0.87     | 0.65     | 0.72     |

### FLOPS
| Environment Samples | Direct Policy (only RL) | Direct Policy (RL after SFT pretaining) | Indirect Policy |
| -------- | -------- | -------- | -------- |
| 0 |  0    |  3.65x10^17    | 6.9x10^12     |
| 128k |  7.36x10^16    | 4.38x10^17     | 4.09×10^14     |
| 256k | 1.46x10^17     | 5.11x10^17     | 8.109×10^14     |
| 1M | 7.12x10^17     | 1.09x10^18     | 3.145x10^15     |


We will add these results in a revised version of paper and extend the comparison to the rest of the domains given enough time to run the experiments.


**Additional baselines and experiments**

To address the reviewers' points, we have conducted several additional experiments. We concisely present these results and conclusions here, leaving detailed discussions in the indivual rebuttals.

- In this [figure](https://imgur.com/a/822YgtT), we reran the LLM Policy baseline by including the LLM-based reward in-context. This did not significantly improve the performance.
- In this [figure](https://imgur.com/a/PWwUJfz), we improve the Reward as Code baseline through environment and LLM feedback.  This approach worked better on MiniWob-Hard, but not on other domains.
- We report results for [direct policy modeling](https://imgur.com/a/vlCNtMV) and [indirect policy modeling](https://imgur.com/a/DMu5dZ1) using a larger set of LLMs/VLMs, bringing the total to 7 models being investigated. The results are generally equivalent across models, except for AI feedback on MetaWorld.
- In this [figure](https://imgur.com/a/2m6odah) we investigate the importance of chain-of-thought for the AI feedback baseline. Removing CoT did not significantly alter the performance.


These additional experiments provide a larger scope and expand the evidence supporting the conclusions of our paper, which remain unchanged.

---

### Meta-Review · Area_Chair_cCrA · 2024-12-23

**Metareview:**

The paper investigates how large language models can be integrated into reinforcement learning workflows, focusing on two approaches: direct policy generation and indirect reward modeling. Through experiments on diverse environments, the authors demonstrate that LLM-based reward modeling using AI feedback consistently outperforms direct policy modeling in both performance and sample efficiency. The paper further explores fine-tuning methods, revealing that fine-tuning for reward modeling mitigates catastrophic forgetting while improving reward quality.

**Strengths**

- The empirical evaluation is comprehensive (R1, R2, R3, R4).
- The empirical observation is interesting (R1, R3, R4).
- The paper is well-organized and easy to follow (R1, R2, R4).

**Weakness**

- Several reviewers note that the paper does not propose a new method, focusing instead on empirical analysis. Some conclusions, such as the benefits of fine-tuning with synthetic data, are unsurprising (R1, R2, R3).
- The handling of inconsistent or low-confidence preferences from LLMs and their sensitivity to subtle state changes remain underexplored (R4).
- The experiments lack detailed statistical analysis of critical phenomena (R2, R3).

The paper lacks significant methodological novelty, though its extensive empirical investigations and practical insights into leveraging LLMs for RL make it a valuable contribution to the field.

**Additional Comments On Reviewer Discussion:**

Novelty (R1, R2, R3): The authors argued that the novelty lies in their comprehensive empirical evaluation and insights into credit assignment and fine-tuning trade-offs. They highlighted several previously unreported observations.

Insufficient related work (R1): The authors discussed the relationship with each of the paper reviewer mentioned.

Statistical analysis, baseline comparisons (R1, R2, R3): The authors conducted additional experiments, including FLOP comparisons for direct and indirect policy modeling, and provided baselines for intrinsic rewards.

Preference Uncertainty (R4): They acknowledged the potential for inconsistent preferences and showed how fine-tuning even on a small dataset can mitigate these issues.

The authors' rebuttal addressed many of the reviewers' concerns, particularly through additional experiments and clarifications. Novelty concerns still remain.

---

### Decision · Program_Chairs · 2025-01-22

Accept (Poster)